# Ultra-Fast Language Generation via Discrete Diffusion Divergence Instruct

**Haoyang Zheng**[1], **Xinyang Liu**[2], **Cindy Xiangrui Kong**[1], **Nan Jiang**[3], **Zheyuan Hu**[4],
**Weijian Luo**[5*], **Wei Deng**[6*], **Guang Lin**[1]
[1]Purdue University, [2]University of Texas at Austin, [3]University of Texas at El Paso,
[4]National University of Singapore, [5]hi-Lab, Xiaohongshu Inc, [6]ML Research, Morgan Stanley

## Abstract

Fast and high-quality language generation is the holy grail that people pursue in the age of AI. In this work, we introduce Discrete Diffusion Divergence Instruct (**DiDi-Instruct**), a training-based method that initializes from a pre-trained diffusion large language model (dLLM) and distills a few-step student for fast generation. The resulting DiDi-Instruct model achieves comparable or superior performance to its dLLM teacher and the GPT-2 baseline while enabling up to $64\times$ acceleration. The theoretical foundation of DiDi-Instruct is a novel framework based on integral KL-divergence minimization, which yields a practical training algorithm. We further introduce *grouped reward normalization*, *intermediate-state matching*, and the *reward-guided ancestral sampler* that significantly improve training stability, model coverage, and inference quality. On OpenWebText, DiDi-Instruct achieves perplexity from 62.2 (8 NFEs) to 18.4 (128 NFEs), which outperforms prior accelerated dLLMs and GPT-2 baseline. These gains come with a negligible entropy loss (around $1\%$) and reduce additional training wall-clock time by more than $20\times$ compared to competing dLLM distillation baselines. We further validate the robustness and effectiveness of DiDi-Instruct through extensive ablation studies, model scaling, downstream tasks, and the generation of discrete protein sequences. In conclusion, DiDi-Instruct is an efficient yet effective distillation method, enabling language generation in the blink of an eye.

**Code:** github.com/haoyangzheng-ai/didi-instruct
**Project Page:** haoyangzheng.github.io/research/didi-instruct/

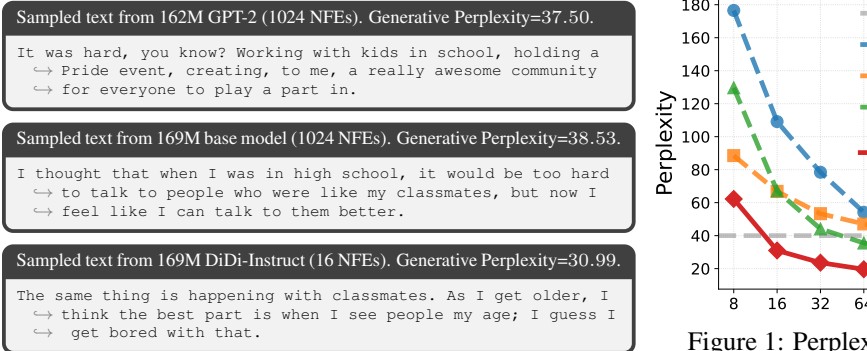

Figure 1: Perplexity vs. NFEs[1].

## 1 Introduction

Fast language sequence generation has been a long-standing goal for large-scale AI systems. Auto-regressive (AR) large language models (LLMs) have achieved remarkable success across a wide range of natural language tasks (Brown et al., 2020; Meta AI, 2023; Achiam et al., 2023; Team

---

*Corresponding authors: pkulwj1994@icloud.com, weideng056@gmail.com.

[1]Baselines include GPT-2, masked diffusion language models (MDLM; Sahoo et al. (2024)), diffusion duality (DUO; Sahoo et al. (2025)), and self-distillation through time (SDTT; Deschenaux & Gulcehre (2025)).

et al., 2024). By predicting the next token from left to right using causal attention of Transformers (Vaswani et al., 2017), AR models can scale to billions or even trillions of parameters while delivering state-of-the-art performance (Touvron et al., 2023). However, the interior mechanism that enables this success also imposes an inherent bottleneck: tokens must be generated sequentially, which limits parallelism and constrains throughput at scales (Leviathan et al., 2023; Kim et al., 2023; Ning et al., 2024). *Even with recent computation optimization, such as KV caches, AR models still face a significant throughput ceiling.*

Inspired by continuous diffusion models for images (Song et al., 2021b; Ho et al., 2020), diffusion large language models (dLLMs) (Li et al., 2022) reinterpret text generation as an iterative denoising process over token sequences, offering a competitive alternative to the AR model. Concretely, a dLLM starts from a fully corrupted sequence and learns to progressively denoise it into a clean text sequence. This inference paradigm leverages bidirectional attention in the Transformer, allowing dLLMs generate sequences with fewer numbers of function evaluations (NFEs) than AR models.

Despite their strong performance and improved efficiency, dLLMs still require many denoising steps during inference, creating an inference time bottleneck. For example, on the OpenWebText benchmark, *dLLMs still need up to 256 NFEs to match the performance of a GPT2 model when generating sequences of length 1024.* To mitigate this limitation, several distillation methods for dLLMs have been proposed to train few-step generators to produce high-quality samples without performance degradation. Notably, Deschenaux & Gulcehre (2025) trains a student to match a teacher's predictions across time-steps, enabling the student to generate at least 32 tokens per step and achieving up to $8\times$ speedups over AR models. Later, Sahoo et al. (2025) relates dLLMs to Gaussian diffusion and introduces curriculum learning and discrete consistency distillation to support few-step sampling. Zhu et al. (2025b) matches the token-level distribution of dLLMs and uses a token initialization strategy to obtain a one-step generator. We defer a detailed, side-by-side introduction and discussions of their distillation mechanisms to Appendix A.3. While these studies advance the frontier of fast language generation, their generation throughput still leaves substantial room for improvement: *on OpenWebText, even the most recent SDTT can not match the performance of GPT2 baseline within 32 NFEs.* Moreover, existing distillation methods are motivated by heuristic design choices, lacking well-established theoretical justification.

In this work, we introduce **Di**screte **Di**ffusion Divergence **Instruct** (**DiDi-Instruct**), a novel distillation framework for fast language generation. The learning objective of DiDi-Instruct is to minimize the Integral Kullback–Leibler (IKL) divergence, introduced by the seminal work of Luo et al. (2023b) for the continuous diffusion model. By minimizing IKL between the distributions of a few-step student generator and a pre-trained teacher dLLM, the student is trained to match the teacher's generation distribution, while achieving substantially improved inference efficiency. Directly adapting IKL to masked diffusion models (MDMs), however, is nontrivial due to the discrete and non-differentiable operations inherent in dLLM generation. To overcome these challenges, we develop a comprehensive solution spanning **objective design**, **training stability**, and **inference efficiency**. Specifically, we make the following contributions:

- *Principled Training for Fast Sequence Generation*: We reformulate the distillation objective from a general policy gradient perspective, deriving a simple yet tractable update rule for the few-step student according to some reward function. Using an adversarial language discriminator to estimate the log-density ratio (reward) between the teacher dLLM and the student, we introduce a practical DiDi-Instruct algorithm that trains the few-step student, paired with an assistant discriminator.
- *Simple yet Effective Techniques in Training and Inference*: We introduce *grouped reward normalization*, *intermediate-state matching*, and the *reward-guided ancestral sampler* (RGAS) that substantially improve the training stability, the model coverage, and the inference performances, reducing the perplexity of generated sequences by 30%.
- *State-of-the-Art Fast Sequence Generation:* DiDi-Instruct achieves new state-of-the-art performance on the OpenWebText benchmark: consistently lower PPL across 8 to 128 NFEs (Figure 1), negligible entropy loss, and over $20\times$ faster distillation; detailed ablations, model scaling, downstream tasks, and protein sequence generation further confirm its robustness.

To the best of our knowledge, DiDi-Instruct is the first framework to successfully apply distribution-matching distillation to MDM-based text generation, along with record-breaking performances in few-step language sequence generation. A discussion of related work is deferred to Appendix A.

## 2 PRELIMINARY

This section introduces the notations for MDMs (Sec. 2.1) and the integral KL divergence (Sec. 2.2), for preparation of the DiDi-Instruct algorithm in Section 3. We consider a vocabulary of size $K$, where each token in the vocabulary is represented by a one-hot vector. The set of all such vectors is $\mathcal{V} = \{\mathbf{x} \in \{0,1\}^K \mid \sum_{j=1}^K x_j = 1\}$. The $K$-th token is reserved for the special [MASK] token, whose one-hot encoding is noted by vector $\mathbf{m}$ with the $K$-th entry being 1 and all others being 0. To generate a length-$L$ sequence $\mathbf{x}^{1:L} = (\mathbf{x}^1, \ldots, \mathbf{x}^L) \in \mathcal{V}^L$, we assume the process factorizes across positions, i.e., each index $\ell \in \{1, \ldots, L\}$ evolves independently. We adopt a continuous time index $t \in [0, 1]$ with a monotonically decreasing noise schedule $\alpha_t \in [0, 1]$ satisfying $\alpha_0 \approx 1$ and $\alpha_1 \approx 0$. Finally, we denote $K$-simplex as $\Delta^K$.

### 2.1 MASKED DIFFUSION MODELS

**Forward process.** In the forward corruption process, once a token transitions to the masked state $\mathbf{m}$, it never reverts. Under independent per-token transitions, $\mathbf{x}^{1:L}$ converges to the fully masked sequence $(\mathbf{m}, \ldots, \mathbf{m})$ with probability 1 as $t \to 1$ (Sahoo et al., 2024). Concretely, a clean token $\mathbf{x} \in \mathcal{V}$ transitions to a latent state $\mathbf{z}_t \in \mathcal{V}$ at time $t \in (0, 1]$. This absorbing-state process implies that $\mathbf{z}_t$ either stays as the original token $\mathbf{x}$ with probability $\alpha_t$ or transitions to $\mathbf{m}$ with probability $1 - \alpha_t$. The forward corruption kernel is:

$$\mathcal{Q}(\mathbf{z}_t \mid \mathbf{x}) = \mathrm{Cat}(\mathbf{z}_t; \alpha_t \mathbf{x} + (1 - \alpha_t)\mathbf{m}), \tag{1}$$

where $\mathrm{Cat}(\mathbf{z}; \boldsymbol{\pi})$ denotes a Categorical distribution over the one-hot vector $\mathbf{z} \in \mathcal{V}$ with probability vector $\boldsymbol{\pi} \in \Delta^K$. For $s < t$, the exact posterior has two cases:

$$\mathcal{Q}(\mathbf{z}_s \mid \mathbf{z}_t, \mathbf{x}) = \begin{cases} \mathrm{Cat}(\mathbf{z}_s; \mathbf{z}_t), & \mathbf{z}_t \neq \mathbf{m}, \\ \mathrm{Cat}\left(\mathbf{z}_s; \dfrac{(1 - \alpha_s)\mathbf{m} + (\alpha_s - \alpha_t)\mathbf{x}}{1 - \alpha_t}\right), & \mathbf{z}_t = \mathbf{m}. \end{cases} \tag{2}$$

The intuition of Equation (2) is: at time $t$, an unmasked token implies a deterministic pash, whereas a masked token considers a probabilistic mixture governed by the noise schedule.

**Backward process.** The reverse process of MDMs iteratively reconstructs the original uncorrupted sequence from the noised input by replacing masked tokens. This is achieved through a learned neural network $\mathbf{p}_\theta : \mathcal{V} \times [0, 1] \to \Delta^K$ that predicts the clean token $\mathbf{x}$ from its corrupted version $\mathbf{z}_t$ at time $t$. The output of $\mathbf{p}_\theta(\mathbf{z}_t, t)$ satisfies that (i) the predicted distribution forms valid categorical distribution that sum to 1, (ii) the prediction $\mathbf{p}_\theta(\mathbf{z}_t, t)$ assigns zero probability mass to the [MASK] token, and (iii) once a token is unmasked, its state remains fixed in all subsequent steps.

When applied to sequences $\mathbf{x}^{1:L}$, the denoising process is assumed to be token-wise independent. A network $\mathbf{p}_\theta^{1:L}(\mathbf{z}_t^{1:L}, t)$ predicts the distribution over the entire sequence, where $\mathbf{p}_\theta^\ell(\mathbf{z}_t^{1:L}, t)$ specifies the distribution for the $\ell$-th token. The reverse transition from $\mathbf{z}_t$ to $\mathbf{z}_s$ is defined as $\mathcal{P}_\theta(\mathbf{z}_s \mid \mathbf{z}_t) = \mathcal{Q}(\mathbf{z}_s \mid \mathbf{z}_t, \mathbf{x} = \mathbf{p}_\theta(\mathbf{z}_t, t))$. We extend this to sequences by factorizing the process over the sequence of length $L$: $\mathcal{P}_\theta(\mathbf{z}_s^{1:L} \mid \mathbf{z}_t^{1:L}) = \prod_{\ell=1}^L \mathcal{Q}(\mathbf{z}_s^\ell \mid \mathbf{z}_t^\ell, \mathbf{p}_\theta^\ell(\mathbf{z}_t^{1:L}, t))$ In continuous time, minimizing the negative evidence lower bound (NELBO) provides a tractable training objective (Shi et al., 2024):

$$\mathcal{L}_{\mathrm{NELBO}}^{\infty^L}(\theta) = \mathbb{E}_q\left[\int_0^1 \frac{\mathrm{d}\alpha_t}{\mathrm{d}t} \frac{1}{1 - \alpha_t}\left[\sum_{\ell : \mathbf{z}_t^\ell = \mathbf{m}} \mathbf{x}^{\ell\top} \log \mathbf{p}_\theta^\ell(\mathbf{z}_t^{1:L}, t)\right] \mathrm{d}t\right]. \tag{3}$$

### 2.2 INTEGRAL KL DIVERGENCE

Let $\mathbf{q}_\theta(\mathbf{z}_\mathbf{t}, t)$ denote the forward teacher marginal and $\mathbf{q}_\nu(\mathbf{z}_\mathbf{t}, t)$ the student marginal parameterized by $\nu$[1]. To distill knowledge from $\mathbf{q}_\theta$ to $\mathbf{q}_\nu$ across different noise levels $t \in [0, 1]$, we minimize the integral Kullback-Leibler (IKL) divergence (introduced in Luo et al. (2023b)), which aggregates the discrepancy between $\mathbf{q}_\nu$ and $\mathbf{q}_\theta$ over the time domain:

$$\mathcal{D}_{\mathrm{IKL}}(\mathbf{q}_\nu \| \mathbf{q}_\theta) := \int_0^1 \omega(t) \mathrm{KL}(\mathbf{q}_\nu \| \mathbf{q}_\theta) \mathrm{d}t = \int_0^1 \omega(t) \mathbb{E}_{\mathbf{z}_t \sim \mathbf{q}_\nu}\left[\log \frac{\mathbf{q}_\nu(\mathbf{z}_t, t)}{\mathbf{q}_\theta(\mathbf{z}_t, t)}\right] \mathrm{d}t, \tag{4}$$

---

[1]Unless otherwise specified, $\mathbf{q}_\theta$ and $\mathbf{q}_\nu$ denote the full collections $\mathbf{q}_\theta^{1:L}(\mathbf{z}_\mathbf{t}, t)$ and $\mathbf{q}_\nu^{1:L}(\mathbf{z}_\mathbf{t}, t)$.

where $\omega(t)$ is a positive weighting function[2]. Intuitively, both $\mathbf{q}_\theta$ and $\mathbf{q}_\nu$ are full-support for any $t \in [0, 1]$ due to diffusion, making each term $\mathrm{KL}(\mathbf{q}_\nu \| \mathbf{q}_\theta)$ finite and smooth. IKL integrates this continuum of reliable comparisons, ensuring the student learns the teacher's complete denoising behavior, which leads to more stable and effective training than only matching the final output.

It is clear that if we can minimize the IKL divergence (4) between a few-step student language model and a pre-trained dLLM teacher model, the student is able to match the teacher's generation ability with much improved efficiency.

## 3 METHODOLOGY

The extension of IKL to distill MDMs presents several inherent challenges. This section details the comprehensive solution via advances in objective design, training stability, and inference efficiency.

### 3.1 DISCRETE DIFFUSION DIVERGENCE INSTRUCTION

A primary challenge in distilling MDMs is the discrete nature of the state space. The gradient formulation in Luo et al. (2023b) relies on differentiating through $\mathbf{z}_t$. In MDMs, the forward process involves non-differentiable operations (e.g., $\arg\max$), making this gradient inapplicable (Zekri & Boullé, 2025). To address this, Zhu et al. (2025b) proposed a proxy model to approximate $\mathbf{z}_t$ and reported promising results for text-to-image tasks. However, its effectiveness in text generation remains underexplored. Instead, we draw inspiration from the policy gradient method (Schulman et al., 2017; Fan et al., 2023) to obtain a rigorous solution. By decomposing the objective gradient as a score function weighted by a log-density ratio, we derive the following gradient estimation.

**Theorem 3.1** (Score-Function Identity). *Let the objective $\mathcal{L}(\nu)$ be the weighted IKL divergences between student and teacher marginals, $\mathbf{q}_\nu$ and $\mathbf{q}_\theta$. The gradient of the objective admits:*

$$\nabla_\nu \mathcal{L}(\nu) = \mathbb{E}_{t \sim \pi(t), \mathbf{x} \sim \mathbf{p}_\nu, \mathbf{z}_t \sim \mathcal{Q}} \left[ \frac{\omega(t)}{\pi(t)} \cdot R(\mathbf{z}_t, t) \cdot \nabla_\nu \log \mathbf{p}_\nu(\mathbf{z}_t = \mathbf{m}, t = 1) \right], \quad (5)$$

*where time $t$ is sampled from distribution $\pi(t)$, and $R(\mathbf{z}_t, t) := \log \mathbf{q}_\nu(\mathbf{z}_t, t) - \log \mathbf{q}_\theta(\mathbf{z}_t, t)$ is the reward (i.e., the log-density ratio between the student and teacher) evaluated at $\mathbf{z}_t$.*

The proof is deferred to Appendix B. Theorem 3.1 shows that the IKL gradient admits a score-function form that does not differentiate through the discrete sampling path. We sample $\mathbf{x} \sim \mathbf{p}_\nu(\mathbf{z}_t = \mathbf{m}, t = 1)$, corrupt it to $\mathbf{z}_t \sim \mathcal{Q}$, and differentiate only $\log \mathbf{p}_\nu(\mathbf{z}_t = \mathbf{m}, t = 1)$. The reward $R(\mathbf{z}_t, t)$ weights the score to guide distillation toward matching the teacher's marginal at $t$.

### 3.2 DENSITY-RATIO ESTIMATION

The reward is a mathematically natural choice for distribution matching. However, both $\log \mathbf{q}_\nu$ and $\log \mathbf{q}_\theta$ are intractable to compute directly. To address this, we avoid estimating the individual densities and instead approximate their ratio. Inspired by Goodfellow et al. (2014); Wang et al. (2025), we train an auxiliary discriminator for this purpose. Intuitively, the optimal discriminator, which differentiates $\mathbf{q}_\nu$ and $\mathbf{q}_\theta$, implicitly encodes their density ratio, as its output probability directly reflects the relative likelihood of the two distributions. We thus model the reward based on the discriminator.

**Lemma 3.2** (Density Ratio Representation). *Let $D_\lambda : \mathcal{V}^L \times [0, 1] \to (0, 1)^L$ be a parameterized discriminator to distinguish samples from the student marginal $\mathbf{q}_\nu$ and the teacher marginal $\mathbf{q}_\theta$. For the optimal discriminator $D_{\lambda^\star}(\mathbf{z}_t, t)$, the density ratio satisfies*

$$\frac{\mathbf{q}_\nu(\mathbf{z}_t, t)}{\mathbf{q}_\theta(\mathbf{z}_t, t)} = \frac{D_{\lambda^\star}(\mathbf{z}_t, t)}{1 - D_{\lambda^\star}(\mathbf{z}_t, t)}.$$

Following Lemma 3.2, we construct a tractable reward signal $R(\mathbf{z}_t, t)$ by aggregating the log-density ratios across all masked positions. Let $M$ denote the number of masked tokens in the sequence. The reward can be estimated using the discriminator $D_\lambda$ as follows:

$$R(\mathbf{z}_t, t) = \frac{1}{M} \sum_{\ell, \mathbf{z}_t^\ell = \mathbf{m}} \log \frac{D_\lambda^\ell(\mathbf{z}_t, t)}{1 - D_\lambda^\ell(\mathbf{z}_t, t)}, \quad (6)$$

---

[2]To simplify notation, we abbreviate $\mathbf{z}_t^{1:L}$ as $\mathbf{z}_t$, and $\mathbf{x}^{1:L}$ as $\mathbf{x}$ unless stated otherwise.

where $D_\lambda^\ell(\cdot, \cdot)$ denotes the $\ell$-th element of $D_\lambda(\cdot, \cdot)$. This formulation provides a tractable and computable reward signal for our objective based on $D_\lambda$. For more details, please check Appendix C.

## 3.3 GROUPED REWARD NORMALIZATION

Although the reward estimator in (6) is tractable, its direct use in score-function gradients can exhibit high variance (Williams, 1992). We therefore adopt Group Relative Policy Optimization (Shao et al., 2024; Team et al., 2025) to standardize rewards within each mini-batch. Given a mini-batch of size $G$, we draw $\{t_i\}_{i=1}^G$ from $\pi(t)$, estimate $\{\mathbf{z}_{t_i}\}_{i=1}^G$ and $\{R(\mathbf{z}_{t_i}, t_i)\}_{i=1}^G$ correspondingly[3]. The final stabilized reward $\widetilde{R}_i$ is obtained by normalizing $R_i$ with the mean $\mu_g$ and variance $\sigma_g^2$:

$$\widetilde{R}_i = \frac{R_i - \mu_g}{\sigma_g + \epsilon}, \qquad \text{where } \mu_g = \tfrac{1}{G} \sum_{i=1}^G R_i \quad \text{and} \quad \sigma_g^2 = \tfrac{1}{G} \sum_{i=1}^G (R_i - \mu_g)^2. \tag{7}$$

Here, a small constant $\epsilon > 0$ is used for numerical stability. Replacing the raw reward $R_i$ with this stabilized reward $\tilde{R}_i$ in the gradient estimator yields a more robust and stable update rule.

## 3.4 THE PROPOSED ALGORITHMS

**Adversarial reward estimation.** We train a parameterized discriminator $D_\lambda$ to separate corrupted samples from $\mathbf{q}_\nu$ and $\mathbf{q}_\theta$ at time $t$. For each mini-batch index $i$, we start from the all-masked sequence to sample $\mathbf{x} \sim \mathbf{p}_\nu$ and $\mathbf{x}' \sim \mathbf{p}_\theta$, then corrupt both to the same noise level $t_i$ via (2) to obtain $\mathbf{z}_i \sim \mathcal{Q}$ and $\mathbf{z}_i' \sim \mathcal{Q}$. The discriminator is optimized with balanced binary cross-entropy:

$$\mathcal{L}_D(\lambda) = -\frac{1}{G} \sum_{i=1}^G \left[ \log D_\lambda(\mathbf{z}_i, t_i) + \log(1 - D_\lambda(\mathbf{z}_i', t_i)) \right]. \tag{8}$$

At optimality (i.e., for $\lambda^\star$), the discriminator satisfies $D_{\lambda^\star}(\mathbf{z}_t, t) \approx \frac{\mathbf{q}_\nu(\mathbf{z}_t, t)}{\mathbf{q}_\nu(\mathbf{z}_t, t) + \mathbf{q}_\theta(\mathbf{z}_t, t)}$. This indicates that $\log \mathrm{it}\, D_{\lambda^\star}$ provides a tractable estimate of the log–density ratio used in (6).

**Remark 3.3.** *Compared with regression-based distillation (e.g., SDTT and DUO), distillation is more delicate to optimize and requires proper regularization, but it is also crucial for achieving high-fidelity generation with very few diffusion steps (Dhariwal & Nichol, 2021). To promote training stability, we (i) initialize both $\mathbf{p}_\nu$ and $D_\lambda$ from the pre-trained teacher, (ii) warm up $D_\lambda$ while freezing $\mathbf{p}_\nu$ for an initial phase, and (iii) clip both rewards and gradients in the policy update to avoid exploding updates. Empirically, we monitor discriminator accuracy and observe that it consistently stays in a non-saturated regime.*

**Score function decomposition.** While sampling directly from $\mathbf{z}_t=\mathbf{m}$ ($t = 1$) to $\mathbf{x}$ is suitable for one-step generators, it induces collapse in multi-step regimes: the student never conditions on intermediate states and tends toward low-entropy, mode-seeking behavior (Zhu et al., 2025b). To expose the student to intermediate corruption levels, we approximate the score from (5) by decomposing it at a randomly sampled time $t_i \sim \pi(t)$ and its corresponding state $\mathbf{z}_i$, which gives:

$$\nabla_\nu \log \mathbf{p}_\nu(\mathbf{z}_t=\mathbf{m}, t = 1) \approx \nabla_\nu \log \mathcal{P}_\nu(\mathbf{z}_i | \mathbf{z}_t=\mathbf{m}) + \nabla_\nu \log \mathbf{p}_\nu(\mathbf{z}_i, t_i). \tag{9}$$

Training with the split score (9) exposes the student to a distribution of intermediate states and mitigates entropy collapse, while remaining compatible with the IKL estimator.

**End-to-end training algorithm.** The training procedure of DiDi-Instruct alternates between updating the discriminator $D_\lambda$ and the student $\mathbf{p}_\nu$, which is summarized in Algorithm 1 and visualized in Figure 2. Each iteration consists of two phases: First, the discriminator is updated to better distinguish between corrupted samples from $\mathbf{p}_\theta$ and $\mathbf{p}_\nu$; We then apply (5) with the normalized reward and the decomposed score to stabilize gradient updates. This alternating scheme yields a robust

---

[3]For clarity and concision, we denote $R(\mathbf{z}_{t_i}^{1:L}, t_i)$ as $R_i$, and $D_\lambda(\mathbf{z}_{t_i}^{1:L}, t_i)$ as $D_i$ throughout.

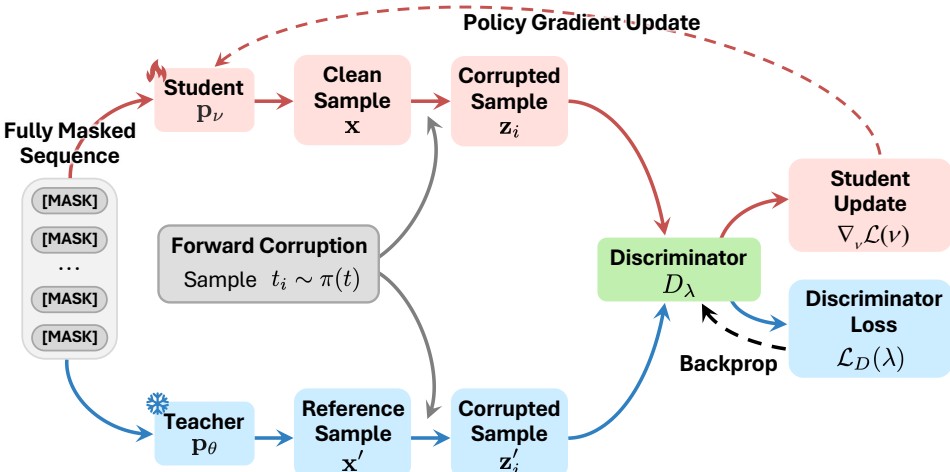

Figure 2: The visualized pipeline of DiDi-Instruct (summarized in Algorithm 1). Given a fully masked input $\mathbf{z}_t$ ($t = 1$), both the student $\mathbf{p}_\nu$ and the teacher (assistant model) $\mathbf{p}_\theta$ produce clean samples $\mathbf{x}$ and $\mathbf{x}'$, which are corrupted at $t_i \sim \pi(t)$ to form $\mathbf{z}_i$ and $\mathbf{z}'_i$. The discriminator $D_\lambda$ is trained to classify these outputs, while its reward signal (6) enables the gradient update (5) for the student. The red line represents the gradient flow for the student's update step, and the blue line represents the gradient flow for the auxiliary model's update step.

few-step student generator that closely matches the teacher's marginals over different corrupt levels.

| **Algorithm 1** DiDi-Instruct Training | **Algorithm 2** RGAS Inference |
|---|---|
| **for** each training step **do**
    Sample $t_i \sim \pi(t)$, and
        $\mathbf{x} \sim \mathbf{p}_\nu(\mathbf{z}_t, t = 1)$, $\mathbf{x}' \sim \mathbf{p}_\theta(\mathbf{z}_t, t = 1)$.
    Corrupt $\mathbf{x}$, $\mathbf{x}'$ to partially masked $\mathbf{z}_i$, $\mathbf{z}'_i$.
    Update discriminator $D_\lambda$ with (8).
    Update student $\mathbf{p}_\nu$ with (5)-(7), and (9).
**return** student $\mathbf{p}_\nu$ and discriminator $D_\lambda$. | Initialize $\mathbf{z}_N \leftarrow (\mathbf{m}, \dots, \mathbf{m})$.
**for** $n = N, \dots, 1$ **do**
    **if** $n$ in early stage **then**
        Sample $\mathbf{z}_{n-1}$ with (10) ($h > 0$, $M = 1$).
    **else**
        Sample $\mathbf{z}_{n-1}$ with (10)-(11) ($h = 0$, $M \geq 1$).
Set $\mathbf{x} = \mathbf{z}_0$ and **return** sequence $\mathbf{x}$. |

**Reward-guided ancestral sampler.** We further propose a decoding strategy that leverages the trained discriminator to guide ancestral sampling (AS) (Shi et al., 2024; Zheng et al., 2025). Starting from a fully masked sequence $\mathbf{z}_N = (\mathbf{m}, \dots, \mathbf{m})$ at $t_N = 1$, the procedure generates samples by iteratively denoising from $t_n$ to $t_{n-1}$ for $n = N, \dots, 1$, following the student's backward distribution $\mathbf{p}_\nu(\mathbf{z}_{n-1}|\mathbf{z}_n)$. Specifically, for $\mathbf{z}_n^\ell \in \mathcal{V}$ ($\ell = 1, \cdots, L$), we consider a tilted transition:

$$
\mathbf{z}_{n-1}^\ell = \begin{cases} \mathbf{z}_n^\ell, & \text{if } \mathbf{z}_n^\ell \neq \mathbf{m}, \\ \sim \text{Cat}\left[\dfrac{(1 - \alpha_{n-1})\mathbf{m} + (\alpha_{n-1} - \alpha_n)\,\mathbf{p}_\nu^\ell(\mathbf{z}_n, t_n)}{1 - \alpha_n}\right], & \text{if } \mathbf{z}_n^\ell = \mathbf{m}, \end{cases} \tag{10}
$$

where the logits for tokens to be unmasked are augmented with the reward gradient:

$$
\mathbf{p}_\nu^\ell(\mathbf{z}_n, t_n) = \begin{cases} \mathbf{e}_{z_n^\ell}, & \text{if } \mathbf{z}_n^\ell \neq \mathbf{m}, \\ \left[\text{softmax}\big(\boldsymbol{\mu}_\nu^\ell(\mathbf{z}_n, t_n) + h\nabla R(\mathbf{z}_n, t_n)\big), 0\right], & \text{if } \mathbf{z}_n^\ell = \mathbf{m}. \end{cases}
$$

Here, $\mathbf{e}_{z_n^\ell} \in \mathcal{V}$ denotes the one-hot vector with a 1 at index $z_n^\ell$ and 0 elsewhere, and $h$ denotes the tilting scale. RGAS adjusts $h$ and the number of candidates $M$ across the denoising process: For early steps ($t_n \approx 1$), we use *gradient tilting* ($h > 0$, $M = 1$) to steer global structure toward high-reward regions. For late steps ($t_n \approx 0$), we switch to *multi-candidate re-ranking* ($h = 0$, $M > 1$): we draw $M$ candidates $\{\mathbf{z}_{n-1}^{(m)}\}_{m=1}^M$[4] from (10) with $h = 0$, then select $\mathbf{z}_{n-1}$ according to:

$$
\mathbf{z}_{n-1} \sim \text{Cat}\left[\dfrac{\exp\big[R(\mathbf{z}_{n-1}^{(m)}, t_{n-1})\big]}{\sum_{m=1}^M \exp\big[R(\mathbf{z}_{n-1}^{(m)}, t_{n-1})\big]}\right]. \tag{11}
$$

---

[4]For simplicity, we denote the $m$-th candidate of $\mathbf{z}_{t_n}^{1:L}$ as $\mathbf{z}_n^{(m)}$.

To decompose score estimation in (9) and sample the intermediate state $\mathbf{z}_i$ at $t_i$, we set $h = 0$ and $M = 1$ to mitigate the risk of reward hacking (Skalse et al., 2022; Gao et al., 2023). The complete procedure is summarized in Algorithm 2.

# 4 EXPERIMENTS

**Experimental setup.** We distill a pre-trained teacher model into an efficient few-step student generator with DiDi-Instruct. All models are trained on OWT (Gokaslan et al., 2019). Following standard practices (Sahoo et al., 2025), we tokenize the corpus using the GPT-2 tokenizer, pack sequences to a context length of 1024, and hold out the last 100,000 documents for validation.

The teacher is a 169M parameter MDLM with a Diffusion Transformer (Peebles & Xie, 2023) (12 layers, 12 attention heads, 768 hidden dimension). We pre-trained this model from scratch for 1024 NFEs, achieving a perplexity of 38.53 and an entropy of 5.22. The student model shares an identical architecture to ensure a fair comparison. The reward model is a 131M parameter network based on the same backbone, but with a new randomly initialized classification head. This head consists of two linear layers with spectral normalization and a SiLU activation function. During distillation, the reward model and the student generator are trained adversarially in an alternating fashion.

The distillation process runs 10,000 iterations using the AdamW optimizer with a learning rate of $10^{-6}$ and no warm-up. The teacher model was pre-trained on 8 NVIDIA H100 GPUs. Subsequently, our DiDi-Instruct distillation is highly efficient, requiring only a single H100 GPU. All training procedures leverage `bfloat16` for acceleration. Additional details are provided in Appendix D.

## 4.1 EXPERIMENTAL RESULT AND ANALYSIS

This section shows that DiDi-Instruct not only achieves state-of-the-art performance in sample quality, particularly in the few-step generation regime, but also offers substantial improvements in training and inference efficiency.

**Generation results.** We evaluate generative quality by sampling text from the distilled student from 8 to 128 NFEs and measuring GPT-2 Large generative PPL and average sequence entropy. Figure 1 shows that DiDi-Instruct consistently outperforms all baselines in PPL across all sampling steps. Notably, with only **16** NFEs, our model's PPL already surpasses that of the 1024-step teacher model. At 1024 NFEs, DiDi-Instruct achieves a final PPL of 15.62, a reduction of over $24\%$ compared to the strongest baseline. These performance gains are achieved with a negligible loss in diversity; the generative entropy from 8 to 128 NFEs is 5.17, 5.21, 5.18, 5.15, and 5.15, respectively (e.g., 1024-step teacher model is 5.22), indicating sample diversity is well-preserved.

We further assess sample fidelity and diversity using MAUVE (Pillutla et al., 2021) and Self-BLEU (Papineni et al., 2002; Montahaei et al., 2019). In appendix E.1, DiDi-Instruct achieves superior distribution matching (higher MAUVE) and reduced mode collapse (lower Self-BLEU) compared to SDTT and DUO, confirming the improvement comes without compromising generative diversity.

**Efficiency-performance tradeoff.** DiDi-Instruct offers substantial computational advantages in both training and inference. Our single-round distillation framework completes training in around one H100 GPU hour, in contrast to the 20+ GPU hours required by multi-round methods (Sahoo et al., 2024; Deschenaux & Gulcehre, 2025). During inference, DiDi-Instruct demonstrates a superior throughput-latency profile. Under a standardized benchmark on a single H100 GPU, it achieves 2366 tokens/sec, representing a $\mathbf{13.2\times}$ speedup over an AR model of the same size at matched perplexity. We provide detailed benchmarks and setup configurations in Appendix E.2.

**Zero-shot likelihood.** Distilling a model for few-step generations can risk degrading its core language understanding and causing mode collapse. We assess this trade-off by evaluating zero-shot PPL on seven out-of-domain corpora (PTB, WikiText, LM1B, LAMBADA, AG News, PubMed, and ArXiv), with results presented in Table 8. The evaluation shows that DiDi-Instruct strikes an effective balance. It consistently improves upon the DUO distilled baseline, validating our superior distillation process. Crucially, while it trails some of the full, undistilled models as expected, it maintains a highly competitive level of performance. These results confirm that DiDi-Instruct achieves its goal of sampling efficiency while preserving robust zero-shot generalization.

**Sample quality.** Appendix E.10 presents full text excerpts with per-sample metrics (PPL and entropy) for the 1024-step teacher and DiDi-Instruct students from 8 to 128 NFEs, which confirms a clear improvement in narrative quality. The 8-step student exhibits expected repetition, a common artifact of rapid sampling. However, this issue is resolved by 16 NFEs, at which point the student model already surpasses the 1024-step teacher in paragraph-level coherence and topic adherence. This ability to construct focused and specific narratives strengthens as NFEs increase to 128, indicating that DiDi-Instruct not only preserves fluency but actively enhances the model's ability to generate structured, coherent text.

## 4.2 ABLATION STUDIES

We conduct comprehensive ablation studies to validate the contribution of each component in DiDi-Instruct. We perform two types of analyses: a cumulative study (Table 1) that progressively adds techniques to a baseline, showing their synergistic benefits, and a leave-one-out study (Table 2) that removes individual components to confirm their necessity. Implementation details can refer to E.5.

Table 1: Cumulative ablation study. We start from a baseline model and *progressively add* the listed tricks on top of the previous row (top→bottom). Metrics are reported as PPL↓ and Entropy↑ for different NFEs. The teacher with 1024 NFEs yields entropy 5.22.

| Configurations | 8 NFEs | | 16 NFEs | | 32 NFEs | | 64 NFEs | | 128 NFEs | |
|---|---|---|---|---|---|---|---|---|---|---|
| | PPL↓ | Entropy↑ | PPL↓ | Entropy↑ | PPL↓ | Entropy↑ | PPL↓ | Entropy↑ | PPL↓ | Entropy ↑ |
| **Baseline (no tricks)** | 803.922 | 5.85 | 311.450 | 5.76 | 174.789 | 5.70 | 113.112 | 5.61 | 96.649 | 5.59 |
| + Score Decompose | 667.830 | 5.83 | 289.720 | 5.76 | 165.809 | 5.70 | 105.880 | 5.61 | 89.350 | 5.59 |
| + Coupled Time $t$ | 101.019 | 5.16 | 75.188 | 5.46 | 48.441 | 5.35 | 35.833 | 5.37 | 30.574 | 5.33 |
| + $\omega(t)$ Correction | 94.955 | 5.21 | 75.607 | 5.22 | 31.651 | 5.20 | 25.271 | 5.16 | 20.980 | 5.12 |
| + $\pi(t)$ Weighting | 92.100 | 5.15 | 43.997 | 5.17 | 32.276 | 5.21 | 26.079 | 5.21 | 21.377 | 5.13 |
| + Regularization | 88.274 | 5.11 | 43.980 | 5.16 | 28.444 | 5.12 | 21.946 | 5.06 | 18.325 | 5.00 |
| + Guided Inference | 62.236 | 5.17 | 38.188 | 5.21 | 24.971 | 5.18 | 21.905 | 5.15 | 18.446 | 5.15 |

Table 2: Leave-one-out ablation study. Each row shows performance *without (w/o)* one trick while keeping the others unchanged. Metrics are PPL↓ and Entropy↑ over different NFEs. The lowest PPL in each NFE column is underlined.

| Configurations | 8 NFEs | | 16 NFEs | | 32 NFEs | | 64 NFEs | | 128 NFEs | |
|---|---|---|---|---|---|---|---|---|---|---|
| | PPL↓ | Entropy↑ | PPL↓ | Entropy↑ | PPL↓ | Entropy↑ | PPL↓ | Entropy↑ | PPL↓ | Entropy ↑ |
| w/o Score Decompose | 33584 | 6.77 | 28962 | 6.77 | 23134 | 6.75 | 14634 | 6.64 | 7983 | 6.51 |
| w/o Coupled Time $t$ | 360.75 | 5.42 | 159.43 | 5.43 | 94.859 | 5.45 | 64.639 | 5.35 | 51.121 | 5.39 |
| w/o $\omega(t)$ Correction | 82.489 | 5.12 | 41.034 | 5.13 | 30.313 | 5.09 | 25.125 | 5.04 | 18.806 | 5.02 |
| w/o $\pi(t)$ Weighting | 69.656 | 5.22 | 40.499 | 5.17 | 25.799 | 5.15 | 21.503 | 5.16 | 19.616 | 5.14 |
| w/o Regularization | 84.594 | 5.20 | 30.994 | 5.22 | 23.603 | 5.20 | 19.609 | 5.18 | 17.499 | 5.17 |
| w/o Guided Inference | 88.274 | 5.11 | 43.980 | 5.16 | 28.444 | 5.12 | 21.946 | 5.06 | 18.325 | 5.00 |
| **Baseline (with all tricks)** | 62.236 | 5.17 | 38.188 | 5.21 | 24.971 | 5.18 | 21.905 | 5.15 | 18.446 | 5.15 |

Our ablation studies reveal distinct roles for each component in our framework. We find that a two-step score decomposition is a non-negotiable cornerstone, providing essential stability without which the model fails to train. The most significant performance gains are driven by coupling $t$ in (5) and (9). Loss shaping with $\omega(t)$ and $\pi(t)$ further smooths optimization (especially around 16 NFEs, while effects at very small/large budgets are modest). Finally, we identify two budget-dependent components: Regularization is crucial for stability at very few NFEs ($\leq 8$ NFEs) but detrimental at higher budgets, while Guided Inference boosts quality at low NFEs and enhances diversity at high NFEs. These highlight a hierarchy of importance and nuanced interactions between the techniques.

To further explore RGAS, we analyze the hyperparameters $h$ and $M$ through performance landscape scans and Functional ANOVA (Hutter et al., 2014). Our analysis reveals that at low NFEs ($< 16$), $h$ dominates performance by guiding global structure; as the computational budget increases (e.g., 32 NFEs), the importance of $M$ rises significantly. More details are presented in Appendices E.5-E.7.

## 4.3 DOWNSTREAM TASK EVALUATION

To validate the practical utility, we evaluate DiDi-Instruct and baselines on downstream tasks. Specifically, we conduct experiments on Domain Adaptation (where we fine-tune models on the MMLU (Hendrycks et al., 2021) and PubMed benchmarks), and Frozen Feature Extraction (using the GLUE MRPC dataset (Dolan & Brockett, 2005)).

On MMLU fine-tuning, DiDi-Instruct achieves the largest reduction in negative log-likelihood after 5,000 fine-tuning steps while matching teacher accuracy. On PubMed, DiDi-Instruct nearly matches the teacher's perplexity while outperforming other distillation methods. Moreover, in frozen-feature evaluation on GLUE MRPC, DiDi-Instruct achieves the best accuracy and F1 score, demonstrating superior semantic representation quality. These results confirm that DiDi-Instruct maintains strong downstream performance while providing substantial efficiency gains. See Appendix E.4 for details.

## 4.4 SCALING UP TEACHER MODEL SIZE

We conduct an incremental scaling study to 424M parameters while keeping the training and inference pipeline identical to the 169M configuration. The teacher model is a pre-trained 424M MDLM, which is a DiT-style transformer with hidden size 1024, 24 blocks, and 16 attention heads (See Table 3 for network architecture details). The student is distilled on a single H100 using the same data, masking policy, optimizer, and schedule. To balance the training, we also scale the reward discriminator to 373M parameters by adopting a deeper network with more trainable parameters.

Figure 4 reports generative PPL and entropy. We report improvements relative to the teacher at matched NFEs. From 8 to 1024 NFEs, DiDi-Instruct yields large and consistent PPL reductions relative to the teacher: 88.5% (8 NFEs), 87.7% (16 NFEs), 85.3% (32 NFEs), 82.7% (64 NFEs), and 79.0% (128 NFEs). Notably, with only 16 NFEs, the distilled model reaches a perplexity of 32.79, marking an 11.4% improvement over the 1024-step teacher baseline. Entropy remains comparable under the same NFE budgets, indicating that diversity is preserved while accuracy improves. Overall, these results confirm that the quality–efficiency advantages of DiDi-Instruct persist at a larger scale with minimal procedural changes. Additional details are provided in Appendix E.8.

## 4.5 ANOTHER APPLICATION: PROTEIN SEQUENCES GENERATION

To demonstrate the applicability of our distillation framework beyond natural language generation, we apply DiDi-Instruct to unconditional protein sequence generation. We adopt the Diffusion Protein Language Model (DPLM) (Wang et al., 2024) , pretrained on UniRef50 (Suzek et al., 2014) with 150M parameters, as the teacher model and distill it into a few-step student generator. Following (Wang et al., 2024), we evaluate sequence quality using the predicted local distance difference test (pLDDT) score, which reflects structural plausibility and foldability. The distilled model retains the teacher's ability to generate variable-length protein sequences while substantially reducing inference cost, as shown in Figure 5.

Our results demonstrate that the distilled student consistently achieves superior pLDDT scores across generation settings ranging from 8 to 512 NFEs. Compared to the teacher model, the student not only preserves the ability to generate variable-length protein sequences but also enhances structural quality in most cases. Moreover, our model surpasses the high-confidence threshold (pLDDT > 70) with as few as 8–32 NFEs, while the teacher requires substantially more NFEs to reach a comparable level. These results highlight that our distillation framework not only improves structural confidence but also delivers stable performance across sequence lengths and generation budgets. Appendix E.9 includes detailed setup, and visual comparisons between low-confidence outputs of DPLM at small NFEs and high-quality samples produced by DiDi-Instruct (Figures 6-7).

To ensure that the observed improvements in structural confidence (pLDDT) do not stem from mode collapse, we evaluate sequence diversity using MMseqs2 clustering (Steinegger & Söding, 2017). Quantitative results confirm that DiDi-Instruct maintains competitive cluster entropy and low cluster sizes comparable to the teacher model, particularly in few-step regimes. This indicates that our method generates diverse, biologically meaningful sequences without collapsing to a few high-confidence modes. Detailed results are provided in Appendix E.9.

## 5 CONCLUSION AND FUTURE WORK

In this work, we introduced DiDi-Instruct, a training-based acceleration framework for fast language generation that distills a high-quality teacher into a few-step student. Our design targets three axes simultaneously: (i) *objective design* via a tractable policy–gradient update driven by a discriminator–estimated reward; (ii) *training stability* through score decomposition and grouped reward

normalization; and (iii) *inference efficiency* through reward-guided ancestral sampling with gradient tilting and re-ranking. Experiments demonstrate strong gains in generation quality, large reductions in training/inference time, and competitive zero-shot generalization, corroborated by comprehensive cumulative and leave-one-out ablations.

We plan to scale DiDi-Instruct to billion-parameter models, which presents a practical challenge due to the memory requirements of concurrently maintaining the teacher, student, and discriminator. Nevertheless, our findings already establish a new state-of-the-art trade-off among comparable methods, with the student model excelling in quality at low NFEs and maintaining diversity at higher computational budgets. We posit that DiDi-Instruct offers a foundational recipe (principled objectives, training stability, and efficient guidance) for developing high-performance generative models.

## ACKNOWLEDGMENTS

We thank the CoreWeave AI cloud platform for supporting part of the computational resources used in this work. Nan Jiang acknowledges support from the Texas Advanced Computing Center (TACC) under award CCR25054. Guang Lin acknowledges support from the National Science Foundation (NSF) under grants DMS-2533878, DMS-2053746, DMS-2134209, ECCS-2328241, CBET-2347401, and OAC-2311848; the U.S. Department of Energy (DOE) Office of Science, Advanced Scientific Computing Research program under award DE-SC0023161; the SciDAC LEADS Institute; and the DOE Fusion Energy Sciences program under grant DE-SC0024583.

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

# Contents

# A RELATED WORKS

## A.1 RELATED WORK ON CONTINUOUS-SPACE DIFFUSION DISTILLATION

Diffusion models (Song et al., 2021b; Ho et al., 2020; Sohl-Dickstein et al., 2015) perturb data by adding Gaussian noise in a forward diffusion process and then learn the reverse-time dynamics (formulated as a stochastic differential equation) using score networks. Early efforts to accelerate sampling reduced the number of calls to iterative samplers using training-free high-order ODE solvers (Song et al., 2021a; Lu et al., 2022; Karras et al., 2022; Xue et al., 2023). However, such solver-based acceleration still suffered from discretization error in less than ten generation steps.

To achieve few-step generation, Luhman & Luhman (2021) and Salimans & Ho (2022) first introduce training the few-step student models by learning the consecutive trajectory mapping of diffusion solvers. Subsequently, the seminal work of Song et al. (2023) and Luo et al. (2023b) opens the one-step generation of diffusion models through different distillation principles. Specifically, consistency models (Song et al., 2023) distill the few-step models with the trajectory consistency principle, resulting in strong performances (Song & Dhariwal, 2024; Lu & Song, 2025; Geng et al., 2025b; Luo et al., 2023a; Kim et al., 2024; Geng et al., 2025a). Diff-Instruct (Luo et al., 2023b) introduces the distribution matching principle that distills one-step generative models, resulting in leading efficient image generative models (Wang et al., 2025; Luo et al., 2025a; 2024; Luo, 2025; Xu et al., 2025; Yin et al., 2024b;a; Xie et al., 2024; Fan et al., 2023; Zhou et al., 2024; Huang et al., 2024; Luo et al., 2025b;c). Later, many other works have also studied the few-step continuous-space generative models from different perspectives (Liu et al., 2023; Geng et al., 2023; Zhou et al., 2025; Nguyen & Tran, 2024; Lin et al., 2024; Boffi et al., 2025; Xu et al., 2024; Xiao et al., 2021; Meng et al., 2023; Zhang & Chen, 2023; Sauer et al., 2023; Ren et al., 2024; Yan et al., 2024; Gu et al., 2024; Chen et al., 2024).

Inspired by Luo et al. (2023b), this work leverages the distribution-matching principle to tackle the more challenging problem of discrete language sequence generation.

## A.2 RELATED WORK ON DISCRETE DIFFUSION MODELS

Early studies of discrete diffusion modeled categorical data via multinomial or argmax-based transitions (Hoogeboom et al., 2021). This line of work was later generalized by D3PM, which introduced structured transition matrices (such as discretized Gaussian kernels, nearest-neighbor transitions, and absorbing states) together with an auxiliary cross-entropy objective (Austin et al., 2021). Continuous-time perspectives and relaxations further clarified the connection between discrete corruption processes and stochastic dynamics (Campbell et al., 2022; Dieleman et al., 2022; Chen et al., 2023). In parallel, diffusion-based ideas were also explored for controllable text generation and sequence-to-sequence modeling (Li et al., 2022; Gong et al., 2023).

For language modeling, two principal families of dLLMs have emerged. MDMs (Lou et al., 2024; Sahoo et al., 2024; Shi et al., 2024; Ou et al., 2025) treat the forward process as progressive masking of tokens. At the highest noise level, every token is replaced by a special `[MASK]` symbol; the reverse process gradually unmask tokens. Once a token is unmasked, it remains fixed, but many tokens can be denoised simultaneously. Intuitively, MDMs essentially train BERT (Devlin et al., 2019) under a hierarchy of noise levels, motivated by a scaling-based rationale for generation (He et al., 2023; Sahoo et al., 2024).Uniform-state diffusion models (USDMs) (Austin et al., 2021; Gulrajani & Hashimoto, 2023; Schiff et al., 2025; Sahoo et al., 2025) instead corrupt tokens by sampling from a uniform distribution over the vocabulary. Consequently, each token can change multiple times during sampling, enabling self-correction and strong theoretical connections to continuous Gaussian diffusion.

Additionally, recent work on dLLMs has advanced along several fronts, including training objectives and theoretical foundations (Lou et al., 2024; Sahoo et al., 2024; Shi et al., 2024; Ou et al., 2025; Zheng et al., 2025), decoding and sampling efficiency (Zheng et al., 2025; Kim et al., 2025; Zhao et al., 2025; Park et al., 2024; Liu et al., 2025), and large-scale model development (Nie et al., 2025; Zhu et al., 2025a; Song et al., 2025; Ye et al., 2025; Xie et al., 2025; Khanna et al., 2025). Together, these studies have substantially improved our understanding of dLLMs and have made diffusion-based language generation increasingly effective and scalable.

Our work is most closely related to this recent line of dLLMs, but differs in focus. Rather than designing a new corruption process or decoding heuristic, we develop a distillation framework for few-step generation based on distribution matching. Inspired by the continuous-time IKL perspective, our method is tailored to models so as to maintain tractable training and inference while improving generation efficiency and quality.

### A.3    RELATED WORK ON FEW-STEP DIFFUSION LARGE LANGUAGE MODELS

To distill dLLMs into few-step generators for fast language generation, a natural approach is to extend the well-developed distillation techniques already established for continuous diffusion models to the discrete domain. However, in MDMs, all tokens are eventually mapped to the masked state, meaning the prior distribution collapses to the fully masked sequence. As a result, generation under MDMs is inherently stochastic, in contrast to the deterministic trajectories available in the continuous case. Consequently, the stochasticity in Masked Diffusion Models (MDMs) arises from the choice of the masking schedule; once a schedule is fixed, the reverse path is deterministic. This stands in sharp contrast to the inherently stochastic reverse dynamics of continuous diffusion. This determinism is problematic for distillation, as training a student model solely on these fixed trajectories risks mode collapse and a failure to capture the teacher's generalization capabilities. This fundamental difference makes the extensive body of work on continuous-state diffusion distillation not directly applicable. To make this distinction concrete, we first show how the following methods address these challenges in discrete diffusion distillation, followed by a comparison of these representative approaches.

**SDTT (Deschenaux & Gulcehre, 2025).**    SDTT progressively distills a multistep teacher diffusion into a fewer-step student model. Due to the absence of an ODE trajectory as in continuous cases, it matches the teacher-student distribution via KL divergence minimization. Similar to DUO (Sahoo et al., 2025), it needs to decrease the number of student steps for training stability gradually. Such a curriculum-based strategy can be inefficient and computationally costly.

**DUO (Sahoo et al., 2025).**    Continuous-space consistency distillation (CD) requires probability flow ODE (PF-ODE) trajectories, while MDMs are inherently stochastic, making direct CD inapplicable. DUO overcomes this by connecting Gaussian diffusion with Uniform-state Diffusion Model (USDM), enabling CD in the discrete setting. By applying an $\arg\max$ to the latent vectors of Gaussian diffusion, continuous vectors are mapped into discrete one-hot tokens. It is proven that the resulting marginal distribution exactly matches that of a USDM under a suitably transformed noise schedule. Consequently, the evolution of a USDM can be described by an ODE, which makes CD feasible. However, performing CD on USDM requires multiple rounds of distillation and a carefully annealed discretization schedule for training stability. Specifically, the CD procedure starts from the USDM diffusion and gradually learns to map two points, $t$ and $t - \Delta t$, along the same ODE trajectory back to the same origin. Initially, since the diffusion model cannot take large steps without losing accuracy, $\Delta t$ must be kept small. Once the model learns to consistently align $t$ and $t - \Delta t$, the step size can be increased, eventually reaching the largest possible step $\Delta t = t$, i.e., the distilled model can jump large steps for few-step inference. A small step is safe but biased, while a large step is unbiased but unstable. This tradeoff is similar to the observation of CD in the continuous domain.

**DiMO (Zhu et al., 2025b).**    DiMO distills multi-step MDMs into a one-step generator. The key idea is to augment the prior distribution: instead of restricting the initial state to a fully-masked sequence, DiMO samples a subset of tokens from the entire vocabulary. This relaxation enriches the prior, allowing the model to leverage partial information during generation and improving both efficiency and diversity of outputs. During training, DiMO employs an on-policy distillation strategy. The one-step generator is supervised to match the teacher's conditional prediction distribution, not its final output distribution. Specifically, it uses the generator's own one-step output to create a pseudo-intermediate state, and then minimizes the divergence between the student's and the teacher's predicted token distributions conditioned on this state. To avoid mode collapse and reduce mismatch with the teacher's training distribution, DiMO introduces a token initialization strategy that mixes mask tokens with random tokens and adds Gaussian perturbations to embeddings.

While both our method and DiMO employ an auxiliary network, its function is fundamentally different. DiMO utilizes an auxiliary model to approximate the intractable gradients of its token-level

distribution matching objective. In contrast, we draw inspiration from policy gradient (Williams, 1992; Schulman et al., 2017; Fan et al., 2023) and use our auxiliary network to evaluate a log-density ratio. This ratio serves as a reward signal, guiding the generator's updates and circumventing the need for direct gradient approximation.

Although DiMO shows promise in text-to-image generation, it should be noted that extending it to our setting reveals a significant modality gap. DiMO is specialized for image codebook tokens characterized by 2D spatial dependencies and visual coherence. This differs fundamentally from pure language generation, which necessitates modeling 1D sequential dependencies and strict syntactic constraints. In our preliminary experiments, a direct adaptation of DiMO yielded comparable generative perplexity but significantly degraded diversity (entropy drops to 4.0), which suggests training instability and potential mode collapse under naive adaptation. Consequently, transferring architectural choices tuned for visual latents to the text domain is non-trivial, and we consider a principled, text-specific adaptation of DiMO to be a valuable area for future work.

**Our advantages over three related methods.** Our work presents a straightforward and intuitive framework for distilling dLLMs by directly matching teacher and student distributions, addressing several key limitations of recent approaches.

Unlike trajectory-based methods (Sahoo et al., 2025; Deschenaux & Gulcehre, 2025) that rely on supervised trajectory matching (forcing the student to mimic the teacher's logits along rigid, predetermined paths), DiDi-Instruct is a direct, single-stage process derived from Integral KL minimization. By reformulating distillation as a policy gradient problem, DiDi-Instruct avoids the heuristic mappings and expensive multi-stage training typical of trajectory-based methods. Crucially, this objective exhibits a natural explore-exploit mechanism: the stochastic sampling of intermediate states $t_i$ and $z_i$ in (9) facilitates exploration of the discrete tokens, while the student's few-step backward update promotes exploitation. This natural mechanism effectively mitigates the mode-collapse issues inherent in trajectory-matching baselines.

Furthermore, by aligning the student with the teacher's entire stochastic process, our framework offers broad applicability to general dLLMs, in contrast to methods tailored for specific architectures, e.g., DUO (Sahoo et al., 2025) relies on USDM for duality. Finally, we provide a mathematically rigorous solution to the problem of non-differentiability in discrete spaces. This avoids the need for biased proxy-gradient estimators, whose performance in purely textual domains remains underexplored (Zhu et al., 2025b), and instead offers a principled path for distillation.

## B    STUDENT OBJECTIVE DERIVATION

Our goal is to train a masked-diffusion *student* whose forward-time marginals match those of a *teacher* across the entire time horizon. The training objective is the IKL divergence between the student and teacher forward marginals, denoted by $\mathbf{q}_\nu$ and $\mathbf{q}_\theta$. In the masked setting, the forward-time corruption kernel is a *forward absorbing process*, e.g.,

$$\mathcal{Q}(\mathbf{z}_t \mid \mathbf{x}) = \mathrm{Cat}\left(\alpha_t \mathbf{x} + (1 - \alpha_t)\mathbf{m}\right),$$

where $\mathbf{m}$ is a fixed masked tokens and $0 \leq \alpha_t \leq 1$. Crucially, this kernel is *independent of* $\nu$: the parameter $\nu$ only enters through the initial student distribution $\mathbf{x} \sim \mathbf{p}_\nu(\mathbf{z}_t = \mathbf{m}, t = 1)$. Nevertheless, the *forward marginal* $\mathbf{q}_\nu(\mathbf{z}_t, t) = \mathbb{E}_{\mathbf{x} \sim \mathbf{p}_\nu}[\mathcal{Q}(\mathbf{z}_t \mid \mathbf{x})]$ still inherits $\nu$-dependence from $\mathbf{p}_\nu$, which is the key to the gradient calculation below.

We write the student and teacher forward-time marginals as

$$\mathbf{q}_\nu(\mathbf{z}_t, t) = \mathbb{E}_{\mathbf{x} \sim \mathbf{p}_\nu}[\mathcal{Q}(\mathbf{z}_t \mid \mathbf{x})]$$
$$\mathbf{q}_\theta(\mathbf{z}_t, t) = \mathbb{E}_{\mathbf{x} \sim \mathbf{p}_\theta}[\mathcal{Q}(\mathbf{z}_t \mid \mathbf{x})].$$

Our goal is to derive a tractable, low-variance gradient for the IKL objective between these two distributions. We here proceed with the detailed derivation of Theorem 3.1.

*Proof of Theorem 3.1.* We begin by formally defining the objective $\mathcal{L}(\nu)$ as the Integral KL divergence. We assume the mask prior $\mathbf{m}$ has full support on the token simplex. Since the teacher uses the same absorbing kernel, $\mathbf{q}_\nu$ and $\mathbf{q}_\theta$ share support (in particular, $\mathrm{supp}(\mathbf{q}_\nu(\mathbf{z}_t, t)) \subseteq \mathrm{supp}(\mathbf{q}_\theta(\mathbf{z}_t, t))$),

so $KL\left(\mathbf{q}_\nu(\mathbf{z}_t,t)\,\middle\|\,\mathbf{q}_\theta(\mathbf{z}_t,t)\right)$ is well-defined and finite for all $t \in [0,1]$. We define the objective $\mathcal{L}(\nu)$ as the Integral KL Divergence between the teacher and the student:

$$
\begin{aligned}
\mathcal{L}(\nu) := D_{KL}(\mathbf{q}_\nu, \mathbf{q}_\theta) &:= \int_0^1 \omega(t) KL\left(\mathbf{q}_\nu(\mathbf{z}_t,t)\,\middle\|\,\mathbf{q}_\theta(\mathbf{z}_t,t)\right)\mathrm{d}t \\
&= \int_0^1 \omega(t)\mathbb{E}_{\mathbf{z}_t\sim\mathbf{q}_\nu}\underbrace{\left[\log\mathbf{q}_\nu(\mathbf{z}_t,t) - \log\mathbf{q}_\theta(\mathbf{z}_t,t)\right]}_{\mathrm{T}_1}\mathrm{d}t
\end{aligned}
\tag{12}
$$

Under mild regularity conditions (bounded $\omega(t)$, dominated convergence, and differentiability of $\mathbf{p}_\nu(\mathbf{z}_t,t)$ w.r.t. $\nu$), we can interchange $\nabla_\theta$ with the expectation and the time integral (and apply Fubini/Tonelli theorem as needed). To evaluate the gradient of $\mathbb{E}_{\mathbf{z}_t\sim\mathbf{q}_\nu}\left[\log\mathbf{q}_\nu(\mathbf{z}_t,t) - \log\mathbf{q}_\theta(\mathbf{z}_t,t)\right]$ w.r.t. $\nu$, we use the identity $\nabla_\theta\mathbb{E}_{y\sim p_\theta}[f(y)] = \mathbb{E}_{y\sim p_\theta}[f(y)\nabla_\theta\log p_\theta(y)] + \mathbb{E}_{y\sim p_\theta}[\nabla_\theta f(y)]$. Applying it to $\mathrm{T}_1$ yields

$$
\begin{aligned}
\nabla_\nu\mathbb{E}_{\mathbf{z}_t\sim\mathbf{q}_\nu}\left[\mathrm{T}_1\right] &= \nabla_\nu\mathbb{E}_{\mathbf{z}_t\sim\mathbf{q}_\nu}\left[\log\mathbf{q}_\nu(\mathbf{z}_t,t) - \log\mathbf{q}_\theta(\mathbf{z}_t,t)]\right] \\
&\overset{(i)}{=} \mathbb{E}_{\mathbf{z}_t\sim\mathbf{q}_\nu}\left[\underbrace{(\log\mathbf{q}_\nu(\mathbf{z}_t,t) - \log\mathbf{q}_\theta(\mathbf{z}_t,t))}_{\mathrm{reward}}\cdot\underbrace{\nabla_\nu\log\mathbf{q}_\nu(\mathbf{z}_t,t)}_{\mathrm{score}}\right] \\
&\quad + \mathbb{E}_{\mathbf{z}_t\sim\mathbf{q}_\nu}\left[\nabla_\nu(\log\mathbf{q}_\nu(\mathbf{z}_t,t) - \log\mathbf{q}_\theta(\mathbf{z}_t,t))\right] \\
&\overset{(ii)}{=} \mathbb{E}_{\mathbf{z}_t\sim\mathbf{q}_\nu}\left[(\log\mathbf{q}_\nu(\mathbf{z}_t,t) - \log\mathbf{q}_\theta(\mathbf{z}_t,t))\cdot\nabla_\nu\log\mathbf{q}_\nu(\mathbf{z}_t,t)\right] \\
&\overset{(iii)}{=} \mathbb{E}_{\mathbf{z}_t\sim\mathbf{q}_\nu}\left[(\log\mathbf{q}_\nu(\mathbf{z}_t,t) - \log\mathbf{q}_\theta(\mathbf{z}_t,t))\cdot\mathbb{E}_{\mathbf{x}\sim\mathbf{p}_\nu}\left[\nabla_\nu\log\mathbf{p}_\nu(\mathbf{z}_t=\mathbf{m},\mathbf{t}=\mathbf{1})\right]\right] \\
&= \mathbb{E}_{\mathbf{x}\sim\mathbf{p}_\nu,\,\mathbf{z}_t\sim\mathcal{Q}}\left[(\log\mathbf{q}_\nu(\mathbf{z}_t,t) - \log\mathbf{q}_\theta(\mathbf{z}_t,t))\cdot\nabla_\nu\log\mathbf{p}_\nu(\mathbf{z}_t=\mathbf{m},\mathbf{t}=\mathbf{1})\right],
\end{aligned}
\tag{13}
$$

where step $(i)$ applies the score-function (log-derivative) identity and is justified by moving $\nabla_\theta$ under the expectation. Step (ii) uses $\nabla_\nu\log\mathbf{q}_\theta(\mathbf{z}_t,t) = 0$ and the fact that

$$
\begin{aligned}
\mathbb{E}_{\mathbf{z}_t\sim\mathbf{q}_\nu}\left[\nabla_\nu\log\mathbf{q}_\nu(\mathbf{z}_t,t)\right] &= \sum_{\mathbf{z}_t}\mathbf{q}_\nu(\mathbf{z}_t,t)\frac{\nabla_\nu\mathbf{q}_\nu(\mathbf{z}_t,t)}{\mathbf{q}_\nu(\mathbf{z}_t)} = \sum_{\mathbf{z}_t}\nabla_\nu\mathbf{q}_\nu(\mathbf{z}_t,t) = \nabla_\nu\sum_{\mathbf{z}_t}\mathbf{q}_\nu(\mathbf{z}_t,t) \\
&= \nabla_\nu(1) = 0.
\end{aligned}
$$

Step $(iii)$ rewrites $\nabla_\nu\log\mathbf{q}_\nu(\mathbf{z}_t,t)$ as $\mathbb{E}_{\mathbf{x}\sim\mathbf{p}_\nu}[\nabla_\nu\log\mathbf{p}_\nu(\mathbf{z}_t=\mathbf{m},t=1)]$. A detailed derivation is as follows:

$$
\begin{aligned}
\nabla_\nu\log\mathbf{q}_\nu(\mathbf{z}_t,t) &= \frac{1}{\mathbf{q}_\nu(\mathbf{z}_t,t)}\nabla_\nu\mathbf{q}_\nu(\mathbf{z}_t,t) \\
&\overset{(i)}{=} \frac{1}{\mathbf{q}_\nu(\mathbf{z}_t,t)}\nabla_\nu\mathbb{E}_{\mathbf{x}\sim\mathbf{p}_\nu}\left[\mathcal{Q}(\mathbf{z}_t\mid\mathbf{x})\right] \\
&\overset{(ii)}{=} \frac{1}{\mathbf{q}_\nu(\mathbf{z}_t,t)}\cdot\mathbb{E}_{\mathbf{x}\sim\mathbf{p}_\nu}\left[\mathcal{Q}(\mathbf{z}_t\mid\mathbf{x})\nabla_\nu\log\mathbf{p}_\nu(\mathbf{z}_t=\mathbf{m},t=1)\right] \\
&= \mathbb{E}_{\mathbf{x}\sim\mathbf{p}_\nu}\left[\frac{\mathcal{Q}(\mathbf{z}_t\mid\mathbf{x})}{\mathbf{q}_\nu(\mathbf{z}_t,t)}\cdot\nabla_\nu\log\mathbf{p}_\nu(\mathbf{z}_t=\mathbf{m},t=1)\right] \\
&\overset{(iii)}{=} \mathbb{E}_{\mathbf{x}\sim\mathbf{p}_\nu}\left[\nabla_\nu\log\mathbf{p}_\nu(\mathbf{z}_t=\mathbf{m},t=1)\right],
\end{aligned}
\tag{14}
$$

where step $(i)$ uses $\mathbf{q}_\nu(\mathbf{z}_t,t) = \mathbb{E}_{\mathbf{x}\sim\mathbf{p}_\nu}[\mathcal{Q}(\mathbf{z}_t\mid\mathbf{x})]$ and pass $\nabla_\nu$ through the expectation by linearity, step $(ii)$ applies $\nabla_\nu\mathbf{p}_\nu(\mathbf{z}_t=\mathbf{m},t=1) = \mathbf{p}_\nu(\mathbf{z}_t=\mathbf{m},t=1)\nabla_\nu\log\mathbf{p}_\nu(\mathbf{z}_t=\mathbf{m},t=1)$ to factor out a log-gradient, and step $(iii)$ follows from Bayes' rule $\mathbf{p}_\nu(\mathbf{z}_t,t) = \mathbf{p}_\nu(\mathbf{x}\mid\mathbf{z}_t) = \frac{\mathcal{Q}(\mathbf{z}_t\mid\mathbf{x})\mathbf{p}_\nu(\mathbf{z}_t=\mathbf{m},t=1)}{\mathbf{q}_\nu(\mathbf{z}_t,t)}$.

We denote $R(\mathbf{z}_t,t) := \log\mathbf{q}_\nu(\mathbf{z}_t,t) - \log\mathbf{q}_\theta(\mathbf{z}_t,t)$. Incorporating (12)-(13), we finally derive the objective:

$$
\nabla_\nu\mathcal{L}(\nu) = \nabla_\nu\int_0^1\omega(t)\mathbb{E}_{\mathbf{z}_t\sim\mathbf{q}_\nu}\left[\log\mathbf{q}_\nu(\mathbf{z}_t,t) - \log\mathbf{q}_\theta(\mathbf{z}_t,t)\right]\mathrm{d}t
$$

$$= \int_0^1 \omega(t) \nabla_\nu \mathbb{E}_{\mathbf{z}_t \sim \mathbf{q}_\nu} \left[ \log \mathbf{q}_\nu(\mathbf{z}_t, t) - \log \mathbf{q}_\theta(\mathbf{z}_t, t) \right] \mathrm{d}t$$

$$\overset{(i)}{=} \int_0^1 \omega(t) \mathbb{E}_{\mathbf{x} \sim \mathbf{p}_\nu, \, \mathbf{z}_t \sim \mathcal{Q}} \left[ (\log \mathbf{q}_\nu(\mathbf{z}_t, t) - \log \mathbf{q}_\theta(\mathbf{z}_t, t)) \cdot \nabla_\nu \log \mathbf{p}_\nu(\mathbf{z}_t = \mathbf{m}, t = 1) \right] \mathrm{d}t,$$

$$= \int_0^1 \omega(t) \mathbb{E}_{\mathbf{x} \sim \mathbf{p}_\nu, \, \mathbf{z}_t \sim \mathcal{Q}} \left[ R(\mathbf{z}_t, t) \cdot \nabla_\nu \log \mathbf{p}_\nu(\mathbf{z}_t = \mathbf{m}, t = 1) \right] \mathrm{d}t, \tag{15}$$

where step $(i)$ substitutes (13) into the time-weighted IKL integral and uses Fubini/Tonelli theorem to swap $\nabla_\nu$ and $\int_0^1 \mathrm{d}t$.

$\square$

**Remark B.1.** *In practice, the objective in (15) is approximated using a Monte Carlo estimator. The integral over time $t$ is replaced with an expectation over a sampling distribution $\pi(t)$, resulting in a single expectation over all random variables $(t, \mathbf{x}, \mathbf{z}_t)$. This unified expectation is then estimated by taking the sample mean over a mini-batch of size $N$:*

$$\nabla_\nu \mathcal{L}(\nu) \approx \frac{1}{N} \sum_{i=1}^{N} \left[ \frac{\omega(t_i)}{\pi(t_i)} \cdot R(\mathbf{z}_{t_i}, t_i) \cdot \nabla_\nu \mathbf{p}_\nu(\mathbf{z}_t = \mathbf{m}, t = 1) \right] \tag{16}$$

*This final expression is the practical Monte Carlo estimator of the full gradient. Conceptually, the gradient is a single expectation over all random variables $(t, \mathbf{x}, \mathbf{z}_t)$, and this estimator approximates that expectation by taking the sample mean over a mini-batch drawn from the respective distributions.*

## C  AUXILIARY DISCRIMINATOR FOR DENSITY RATIO ESTIMATION

Our student objective, as derived in Appendix B, relies on the reward term $R(\mathbf{z}_t, t) := \log \mathbf{q}_\nu(\mathbf{z}_t, t) - \log \mathbf{q}_\theta(\mathbf{z}_t, t)$, which is intractable due to the unknown marginal distributions $\mathbf{q}_\nu$ and $\mathbf{q}_\theta$. To overcome this, we introduce a tractable estimator for the density ratio $\frac{\mathbf{q}_\nu(\mathbf{z}_t)}{\mathbf{q}_\theta(\mathbf{z}_t)}$ by training an auxiliary discriminator network. This approach is inspired by the principles of Generative Adversarial Networks (GANs) (Goodfellow et al., 2014; Wang et al., 2023) and has been successfully applied in Wang et al. (2025). We hereby adapt the following lemma to MDM setting.

For clarity, we adopt a time-agnostic notation where the corruption level is parameterized by the masking ratio $t \in [0, 1]$, which corresponds to the schedule $\alpha_t$ via $t = 1 - \alpha_t$. Let $\mathbf{z}_t \in \mathcal{V}^L$ denote a sequence with a mask ratio at time $t$.

**Lemma C.1** (Density Ratio Representation). *Let $\mathbf{q}_\theta(\mathbf{z}_t, t)$ and $\mathbf{q}_\nu(\mathbf{z}_t, t)$ be the teacher and student models respectively, over the discrete state space $\mathcal{V}^L$ at a mask ratio $t$. Consider a discriminator $D_\lambda : \mathcal{V}^L \times [0, 1] \to (0, 1)^L$, which outputs a probability for each position $\ell \in \{1, \ldots, L\}$. The discriminator $D_\lambda$ is trained to minimize the objective*

$$\mathcal{L}_D(\lambda) = \frac{1}{L} \sum_{\ell=1}^{L} \mathbb{E}_{\mathbf{z}_t \sim \mathbf{q}_\nu} [-\log D_\lambda(\mathbf{z}_t, t)] + \mathbb{E}_{\mathbf{z}_t \sim \mathbf{q}_\theta} [-\log(1 - D_\lambda(\mathbf{z}_t, t))] \tag{17}$$

*The unique optimal discriminator $D_{\lambda^\star}(\mathbf{z}_t, t)$ that minimizes this objective is given by:*

$$D_{\lambda^\star}(\mathbf{z}_t, t) = \frac{\mathbf{q}_\theta(\mathbf{z}_t, t)}{\mathbf{q}_\theta(\mathbf{z}_t, t) + \mathbf{q}_\nu(\mathbf{z}_t, t)}$$

*Consequently, the density ratio can be expressed directly in terms of the optimal discriminator's output:*

$$\frac{\mathbf{q}_\nu(\mathbf{z}_t, t)}{\mathbf{q}_\theta(\mathbf{z}_t, t)} = \frac{D_{\lambda^\star}(\mathbf{z}_t, t)}{1 - D_{\lambda^\star}(\mathbf{z}_t, t)}.$$

*Proof.* We here provide a detailed derivation for the optimal discriminator in the context of our discrete state space. The objective $\mathcal{J}(D)$ is an expectation over the discrete random variable $\mathbf{z}_t$. We can express this expectation as a summation over all possible sequences $\mathbf{z}_t \in \mathcal{V}^L$:

$$\mathcal{J}(D) = \sum_{\mathbf{z}_t \in \mathcal{V}^L} [-\mathbf{q}_\nu(\mathbf{z}_t, t) \log D_\lambda(\mathbf{z}_t, t) - \mathbf{q}_\theta(\mathbf{z}_t, t) \log(1 - D_\lambda(\mathbf{z}_t, t))]$$

The loss is a sum of terms, where each term depends only on the value of $D_\lambda(\mathbf{z}_t, t)$ for a specific sequence $\mathbf{z}_t$. Therefore, we can find the optimal discriminator $D_{\lambda^\star}(\mathbf{z}_t, t)$ by minimizing the summand pointwise for each $\mathbf{z}_t \in \mathcal{V}^L$ independently. For an arbitrary sequence $\mathbf{z}_t'$, we find the optimal value $D_\lambda(\mathbf{z}_t', t)$ by taking the derivative of the summand with respect to $D_\lambda(\mathbf{z}_t', t)$ and setting it to zero.

$$
\frac{\partial}{\partial D_\lambda(\mathbf{z}_t', t)} \left[ -\mathbf{q}_\nu(\mathbf{z}_t', t') \log D_\lambda(\mathbf{z}_t', t) - \mathbf{q}_\theta(\mathbf{z}_t', t') \log(1 - D_\lambda(\mathbf{z}_t', t)) \right]
$$
$$
\overset{(i)}{=} -\frac{\mathbf{q}_\nu(\mathbf{z}_t', t')}{D_\lambda(\mathbf{z}_t', t)} + \frac{\mathbf{q}_\theta(\mathbf{z}_t', t')}{1 - D_\lambda(\mathbf{z}_t', t)}
$$

(18)

where step $(i)$ follows from standard differentiation of the logarithm function. We set (18) to be zero and get:

$$
-\frac{\mathbf{q}_\nu(\mathbf{z}_t', t')}{D_\lambda(\mathbf{z}_t', t)} + \frac{\mathbf{q}_\theta(\mathbf{z}_t', t')}{1 - D_\lambda(\mathbf{z}_t', t)} = 0
$$
$$
\implies (1 - D_\lambda(\mathbf{z}_t', t))\mathbf{q}_\nu(\mathbf{z}_t', t') = D_\lambda(\mathbf{z}_t', t)\mathbf{q}_\theta(\mathbf{z}_t', t')
$$
(19)
$$
\implies D_{\lambda^\star}(\mathbf{z}_t, t) = \frac{\mathbf{q}_\nu(\mathbf{z}_t', t')}{\mathbf{q}_\nu(\mathbf{z}_t', t') + \mathbf{q}_\theta(\mathbf{z}_t', t')},
$$

which represents an algebraic rearrangement to solve for $D_\lambda(\mathbf{z}_t', t)$. Since this holds for any arbitrary sequence $\mathbf{z}_t'$, we have established the general form of the optimal discriminator $D_{\lambda^\star}(\mathbf{z}_t, t)$.

Finally, we rearrange the expression for $D_{\lambda^\star}(\mathbf{z}_t, t)$ to recover the density ratio:

$$
D_{\lambda^\star}(\mathbf{z}_t, t) \left( \mathbf{q}_\nu(\mathbf{z}_t, t) + \mathbf{q}_\theta(\mathbf{z}_t, t) \right) = \mathbf{q}_\nu(\mathbf{z}_t, t)
$$
$$
\implies D_{\lambda^\star}(\mathbf{z}_t, t)\mathbf{q}_\theta(\mathbf{z}_t, t) = \mathbf{q}_\nu(\mathbf{z}_t, t) \left( 1 - D_{\lambda^\star}(\mathbf{z}_t, t) \right)
$$
$$
\implies \frac{\mathbf{q}_\nu(\mathbf{z}_t, t)}{\mathbf{q}_\theta(\mathbf{z}_t, t)} = \frac{D_{\lambda^\star}(\mathbf{z}_t, t)}{1 - D_{\lambda^\star}(\mathbf{z}_t, t)}.
$$

The point-wise nature of the derivation ensures its validity for discrete probability mass functions. $\square$

**Remark C.2.** *The proof confirms that Lemma C.1 is robust to the specific properties of our MDM framework, including the discrete state space and the use of a mask ratio $t$ as the conditioning variable instead of a continuous time $t$. By training an auxiliary discriminator network $D_\lambda$ to approximate $D^*$, we obtain a tractable, principled estimator for the density ratio, which in turn provides a computable reward signal*

$$
\hat{R}(\mathbf{z}_t, t) = \frac{1}{M} \sum_{\ell, \mathbf{z}_t^\ell = \mathbf{m}} \log \left( \frac{D_\lambda^\ell(\mathbf{z}_t, t)}{1 - D_\lambda^\ell(\mathbf{z}_t, t)} \right)
$$

*for our student objective, where $M$ denotes the number of masked tokens in the sequence.*

# D  ADDITIONAL EXPERIMENTAL SETUP

## D.1  BASELINE SETUP

To ensure a fair and reproducible comparison, we strictly follow the official implementations and experimental configurations of both SDTT and DUO. For SDTT (Deschenaux & Gulcehre, 2025), we adopt the same 169M MDM teacher used in our method, pretrained for 1024 NFEs, and perform seven rounds of distillation with 10K steps each (70K steps in total) to obtain an 8-step student, exactly matching the protocol in the official repository. For DUO (Sahoo et al., 2025), we use the 169M USDM teacher (1024 NFEs) released in their codebase and conduct five rounds of distillation (50K total steps). We additionally evaluate their pretrained few-step student model included in the repository.

In practice, we observe non-negligible variation in the performance of these baselines across different training runs. To provide the most stable and equitable comparison, we report the metrics published in their original papers, which also match the strongest results we reproduced internally.

## D.2 MODEL ARCHITECTURE

Our model architectures are based on the MDLM setting (Sahoo et al., 2024), which utilizes a Diffusion Transformer (Peebles & Xie, 2023) with Rotary Position Embeddings (RoPE) (Su et al., 2024). We conduct experiments at two different scales. The first is a 169M parameter setting, and the second is a larger 424M parameter setting. The detailed hyperparameters for each are listed in Table 3 and Table 4, respectively.

Table 3: Model Architecture Specifications (169M).

| Hyperparameter | Teacher/Student | Discriminator Model |
|---|---|---|
| Model Type | MDLM | MDLM |
| Total Parameters | 169M | 131M |
| Num Layers | 12 | 12 |
| Hidden Size | 768 | 768 |
| Num Attention Heads | 12 | 12 |
| Positional Encoding | RoPE | RoPE |
| Context Length | 1024 | 1024 |
| Vocabulary Size | 50257 | 50257 |
| Classification Head | - | 2 linear layers (SpecNorm) |

Table 4: Model Architecture Specifications (424M).

| Hyperparameter | Teacher/Student | Discriminator Model |
|---|---|---|
| Model Type | MDLM | MDLM |
| Total Parameters | 424M | 373M |
| Num Layers | 24 | 24 |
| Hidden Size | 1024 | 1024 |
| Num Attention Heads | 16 | 16 |
| Positional Encoding | RoPE | RoPE |
| Context Length | 1024 | 1024 |
| Vocabulary Size | 50257 | 50257 |
| Classification Head | - | 2 linear layers (SpecNorm) |

## D.3 DATASET DETAILS

We use the OpenWebText corpus for experiments. The raw text is tokenized using the standard GPT-2 tokenizer. Following Sahoo et al. (2025), we concatenate all documents and pack them into fixed-length sequences of 1024 tokens, adding an <|endoftext|> token between documents. The final 100,000 documents of the corpus are reserved as the validation set.

For zero-shot evaluation, we assess the model's generalization on the following seven diverse, out-of-domain datasets to measure its broader language understanding capabilities.

**PTB:** The Penn Treebank dataset is a widely used benchmark in natural language processing, composed of articles from the Wall Street Journal.

**Wikitext:** The Wikitext-103 dataset is a large corpus of high-quality, long-form articles extracted from Wikipedia, serving as a standard for language modeling.

**LM1B:** The One Billion Word Benchmark is a large-scale dataset derived from a news crawl, often used for training and evaluating large language models.

**Lambada:** The LAMBADA dataset is designed to evaluate a model's ability to comprehend long-range dependencies in text, requiring the prediction of the final word in a passage.

**AG News:** The AG News dataset is a collection of news articles classified into four categories. For our experiments, we use the raw text for perplexity evaluation.

**Pubmed:** This dataset consists of abstracts from biomedical literature, representing a specialized scientific domain.

**Arxiv:** This dataset comprises scientific preprints from various quantitative fields available on the arXiv server, offering another domain of specialized, formal text.

Table 5: Pre-Training and Distillation Hyperparameters.

| Hyperparameter | Pre-training (Teacher) | Distillation (Student) |
|---|---|---|
| Optimizer | AdamW | AdamW |
| Learning Rate | $1.0 \times 10^{-4}$ | $1.0 \times 10^{-6}$ |
| LR Schedule | Cosine Decay | Linear Decay |
| Warm-up Steps | 2,500 | 0 |
| Decay Rate $\beta_1$ | 0.9 | 0.9 |
| Decay Rate $\beta_2$ | 0.95 | 0.999 |
| Weight Decay | 0.05 | 0.01 |
| Global Batch Size | 336 (42 per GPU) | 8 |
| Training Iterations | 200,000 | 10,000 |
| EMA Decay | 0.9999 | 0.9999 |
| **DiDi-Instruct Specific Parameters** | | |
| $\omega(t)$ weight schedules | We test linear schedule $\omega(t) = 1$ and $\omega(t) = -\alpha_t/(1 - \alpha_t + 10^{-8})$. | |
| $\pi(t)$ sampling schedules | We test linear schedule Beta(1, 1), Beta(2, 5), and Beta(5, 2). | |

### D.4 TRAINING AND DISTILLATION HYPERPARAMETERS

The detailed hyperparameters for teacher model pre-training and student model distillation are provided in Table 5. For our 169M models, we pre-trained the teacher model for around 200,000 steps. The subsequent distillation process was completed in about one hour on a single H100 GPU. For the 424M models, the teacher was pre-trained for around 350,000 steps, while the corresponding distillation took approximately two hours on a single H100 GPU.

### D.5 EVALUATION DETAILS

**Generative perplexity:** We use the implementation from Hugging Face's `transformers` library to compute the perplexity of generated samples under a frozen GPT-2 Large model. Text is processed identically to the training data.

**Entropy:** To assess sample diversity, we generated 40 samples for each configuration. Each sample has a fixed length of 1024 tokens. We then computed the average token-level entropy across all generated samples.

**Floating-point precision:** All sampling and evaluation procedures were conducted using `float64` precision. This is to avoid potential artifacts and misleading scores associated with lower-precision arithmetic, a practice highlighted as important for dLLMs (Sahoo et al., 2025).

## E ADDITIONAL EXPERIMENTAL RESULTS

### E.1 DIVERSITY AND FIDELITY ANALYSIS

To provide a comprehensive comparison beyond perplexity, we evaluate sample diversity using MAUVE (for distribution similarity) and Self-BLEU (for diversity). All models generate 1,024 unconditional samples with identical decoding configurations (no prompt, max length 1024). Baselines use the same sampling budgets (8, 16, 32, 64, and 128 NFEs), and all samples are processed using the same GPT-2 tokenizer.

**MAUVE** measures the similarity between the generated distribution $Q$ and the reference distribution $P$ by analyzing the trade-off between Type I and Type II errors. Let $R_\lambda = \lambda P + (1 - \lambda)Q$ be a

mixture distribution for $\lambda \in (0, 1)$. We compute the divergence frontier as a curve $(x(\lambda), y(\lambda))$ in the 2D plane, where the coordinates are exponentially scaled Kullback-Leibler (KL) divergences:

$$x(\lambda) = \exp\left(-c \cdot D_{\mathrm{KL}}(Q\|R_\lambda)\right) \quad \text{and} \quad y(\lambda) = \exp\left(-c \cdot D_{\mathrm{KL}}(P\|R_\lambda)\right), \qquad (20)$$

where $c > 0$ is a scaling factor (we set $c = 5$). As $\lambda$ varies, these coordinates trace a curve within the unit square $[0, 1]^2$. The MAUVE score is calculated as the area under this curve using numerical integration (trapezoidal rule) over a discretized set of $\lambda$ values. Normally, texts are truncated to 256 tokens before embedding.

**Self-BLEU** evaluates diversity through n-gram overlap. For $N$ samples $\{\mathbf{x}_i\}_{i=1}^N$:

$$\text{Self-BLEU} = \frac{1}{N}\sum_{i=1}^N \text{BLEU}(\mathbf{x}_i, \{\mathbf{x}_j\}_{j\neq i}), \qquad \text{where BLEU} = BP \cdot \exp\left(\sum_{n=1}^5 w_n \log p_n\right),$$

where $p_n$ is the n-gram precision, $w_n = 0.2$, and $BP$ is the brevity penalty. We set $n = 5$, use the smoothing method 1, and the tokenization of NLTK.

Table 6: Additional generation metrics: MAUVE ($\uparrow$, higher is better) and Self-BLEU ($\downarrow$, lower indicates more diversity). Comparison across varying NFEs.

| Metrics | Methods | 8 NFEs | 16 NFEs | 32 NFEs | 64 NFEs | 128 NFEs |
|---|---|---|---|---|---|---|
| MAUVE ($\uparrow$) | MDLM | 4.77 | 6.76 | 6.46 | 5.50 | 5.97 |
| ($\times 10^{-3}$) | SDTT | 5.60 | 5.38 | 5.70 | 5.04 | 5.96 |
| | DUO | 5.75 | 5.15 | 5.99 | 5.85 | 6.05 |
| | **DiDi-Instruct (Ours)** | 5.84 | 6.55 | 6.54 | 6.14 | 6.31 |
| Self-BLEU ($\downarrow$) | MDLM | 2.91 | 4.58 | 6.36 | 7.97 | 9.33 |
| ($\times 10^{-2}$) | SDTT | 5.02 | 7.98 | 10.09 | 12.49 | 12.92 |
| | DUO | 8.26 | 9.28 | 9.62 | 8.84 | 8.94 |
| | **DiDi-Instruct (Ours)** | 4.82 | 4.64 | 5.24 | 6.25 | 8.24 |

Table 6 presents the full results. DiDi-Instruct consistently achieves the highest MAUVE scores across all NFEs, which validates that our adversarial distribution-matching objective successfully aligns the student's marginals with the teacher's. Self-BLEU values remain competitive, demonstrating that diversity is not sacrificed for fidelity. Intuitively, baselines such as SDTT and DUO tend to prioritize pointwise accuracy, leading to mode-seeking behavior and worse Self-BLEU. In contrast, score decomposition forces the student to model noisy state evolution, preserving diverse modes, while adversarial training ensures marginal distribution matching for higher fidelity.

### E.2 PRACTICAL LATENCY AND THROUGHPUT ANALYSIS

To complement the NFE-based analysis in the main text, we conduct a practical latency evaluation. All experiments use a single H100 GPU with `bfloat16` precision, batch size 16, and sequence length 1024. All models share the same architecture as demonstrated in Table 3. The AR baseline uses greedy decoding with standard optimizations such as KV-caching and FlashAttention.

Table 7: Practical latency and throughput comparison on one H100 GPU.

| Models | NFEs | Tokens/sec | Latency per 1K tokens (sec) |
|---|---|---|---|
| AR | 1024 | 179.042 | 5.719 |
| MDLM | 512 | 111.146 | 9.213 |
| SDTT | 64 | 806.498 | 1.269 |
| **DiDi-Instruct (Ours)** | **16** | **2366.604** | **0.432** |

The latency across different methods is reported in Table 7. At matched perplexity, DiDi-Instruct achieves a $13.2\times$ reduction in latency per 1k tokens relative to AR, while maintaining high sample quality. This confirms that the advantages of distilled diffusion models persist under realistic

decoding settings and not merely under NFE-based comparisons. Note that the measured latency improvement is a conservative estimate: DiDi-Instruct is fully compatible with hybrid AR–diffusion architectures (Arriola et al., 2025), and benefits from recent advances enabling KV-cache reuse and parallel diffusion decoding (Wu et al., 2026), which offer additional opportunities for further acceleration.

### E.3 RESULTS ON ZERO-SHOT PERPLEXITIES

As detailed in Table 8, DiDi-Instruct demonstrates superior generalization capabilities compared to the distilled baseline. It achieves lower perplexity across different datasets relative to DUO distilled. On benchmarks like Wikitext, it notably surpasses the autoregressive GPT-2 baseline. These results confirm that DiDi-Instruct effectively preserves the teacher's semantic coverage, mitigating the mode collapse typically associated with few-step diffusion generation.

Table 8: Evaluation of zero-shot generalization on out-of-domain datasets. We report PPL to assess whether DiDi-Instruct preserves general language understanding after being distilled for sampling efficiency. This test is crucial to verify that the model avoids mode collapse. Lower PPL ($\downarrow$) indicates better performance. All models are of comparable 170M parameter size, and perplexities for diffusion models are variational upper bounds.

|  | Models | PTB | Wikitext | LM1B | Lambada | AG News | Pubmed | Arxiv |
|---|---|---|---|---|---|---|---|---|
| AR | GPT2[†‡] | 82.05 | 41.60 | 51.25 | 45.04 | 52.09 | 49.01 | 41.73 |
| MDMs | SEDD[†‡] | 100.09 | 40.62 | 68.20 | 50.92 | 62.09 | 44.53 | 38.48 |
|  | MDLM[‡] | 95.26 | 32.83 | 67.01 | 47.52 | 61.15 | 41.89 | 37.37 |
| USDMs | SEDD[†¶] | 105.51 | 49.60 | 82.62 | 65.40 | 82.64 | 55.89 | 50.86 |
|  | UDLM[¶] | 112.82 | 39.42 | 77.59 | 53.57 | 80.96 | 50.98 | 44.08 |
|  | DUO base[¶] | 89.35 | 33.57 | 73.86 | 49.78 | 67.81 | 44.48 | 40.39 |
| Distilled | DUO distilled[§] | 112.27 | 42.53 | 89.45 | 61.17 | 88.70 | 56.17 | 50.27 |
|  | **DiDi-Instruct (Ours)** | 107.03 | 35.20 | 80.58 | 53.62 | 78.36 | 47.56 | 45.35 |

### E.4 DOWNSTREAM TASK EVALUATION

We conduct three downstream evaluations used to verify whether DiDi-Instruct preserves the task capability of the teacher's semantic representations while enabling few-step generation. All experiments use the same 169M architecture unless otherwise specified.

**Domain adaptation on MMLU.** We fine-tune MDLM, SDTT, and DiDi-Instruct on the MMLU benchmark for 5,000 supervised fine-tuning (SFT) steps. Training uses the AdamW optimizer with a base learning rate of $10^{-5}$, which is linearly warmed up for the first 2,500 steps and kept constant thereafter. We employ a batch size of 32 and an evaluation batch size of 4. As shown in Table 9, DiDi-Instruct not only matches the teacher's downstream accuracy but also achieves the most significant improvement in Negative Log-Likelihood ($\Delta$NLL). This indicates robust adaptability to general knowledge domains.

Table 9: Domain adaptation on MMLU dataset.

| Models | NLL (pre) | NLL (post) $\downarrow$ | $\Delta$NLL $\uparrow$ | Acc (pre) | Acc (post) $\uparrow$ | $\Delta$Acc $\uparrow$ |
|---|---|---|---|---|---|---|
| MDLM | 9.879 | 2.814 | 7.065 | 20.6% | 25.3% | 4.7% |
| SDTT | 9.773 | 2.844 | 6.929 | 21.2% | 24.3% | 3.1% |
| **DiDi-Instruct (Ours)** | 9.960 | 2.815 | 7.145 | 21.6% | 25.1% | 3.5% |

---

[3]Results are compiled from prior work for a comprehensive comparison.
[†]Values from Lou et al. (2024).
[‡]Values from the evaluation suite in Sahoo et al. (2024).
[¶]Values reported in Sahoo et al. (2025).
[§]Result obtained by evaluating the DUO distilled model from Sahoo et al. (2025).

**Scientific domain adaptation.** We further evaluate domain transfer to scientific text by conducting 5,000 SFT steps on the PubMed corpus. All models are fine-tuned using AdamW with a learning rate of $5 \times 10^{-6}$ and a constant schedule with 1,000 warmup steps. Training uses a per-GPU batch size of 36 with gradient accumulation (effective batch size 72). As reported in Table 10, DiDi-Instruct significantly outperforms the SDTT and DUO baselines. Notably, it achieves a final perplexity of 24.894, which is practically indistinguishable from the teacher's perplexity, which validates its effectiveness in specialized domains.

Table 10: Domain adaptation on PubMed dataset.

| Models | NLL (pre) | PPL (pre) | NLL (post) $\downarrow$ | PPL (post) $\downarrow$ |
|---|---|---|---|---|
| MDLM | 3.761 | 42.983 | 3.217 | 24.944 |
| SDTT | 3.938 | 51.336 | 3.243 | 25.602 |
| DUO | 4.028 | 56.174 | 3.338 | 27.734 |
| **DiDi-Instruct (Ours )** | 3.879 | 47.563 | 3.215 | 24.894 |

**Frozen feature extractor evaluation.** To assess the quality of the learned latent representations, we evaluate the models on the GLUE MRPC task (Dolan & Brockett, 2005). The pretrained backbones (MDLM, SDTT, DiDi-Instruct) are kept fixed, and a lightweight classification head is added on top of the encoder's pooled hidden representation. The head consists of a single linear layer mapping the pooled vector to two logits, and we additionally unfreeze the final encoder block to allow light adaptation. Training uses AdamW with a learning rate of $2 \times 10^{-5}$, weight decay 0.01, a batch size of 8, and 10,000 training steps. We report Accuracy and F1 on the validation split.

Results in Table 11 show that DiDi-Instruct attains the highest Accuracy and F1 scores. This confirms that DiDi-Instruct preserves and potentially improves semantic discriminability.

Table 11: Frozen feature evaluation on GLUE MRPC dataset.

| Models | Accuracy $\uparrow$ | F1 Score $\uparrow$ |
|---|---|---|
| MDLM | 59.6% | 72.2% |
| SDTT | 59.1% | 72.0% |
| **DiDi-Instruct (Ours)** | 62.3% | 73.3% |

Across all downstream tasks, DiDi-Instruct matches or exceeds the teacher's performance, establishing that few-step distillation maintains strong semantic representation quality while delivering significant inference speedups.

### E.5 IMPLEMENTATION DETAILS OF THE ABLATION STUDIES

To provide a clear basis for interpreting our experimental results, we now elaborate on the precise implementation of each component tested in our ablation studies. This includes the mechanisms and key hyperparameters that were modified between the baseline and full models. Below, we detail the function of each component in our ablation studies.

**Score decompose.** The baseline model is a one-step generator, mapping a fully masked input $\mathbf{z}_t$ ($t = 1$) directly to the final output $\mathbf{x}$. With this technique, we decompose the generation into a two-step process $\mathbf{z}_t \rightarrow \mathbf{z}_\tau \rightarrow \mathbf{x}$ where $t = 1$ and $\tau \in (0, 1)$. The model first generates an intermediate state $\mathbf{z}_\tau$ at a randomly sampled time following $\pi(t)$ and then generates the final output $\mathbf{x}$ from $\mathbf{z}_\tau$. The ablation ('w/o Score Decompose') reverts to the direct, single-step generation.

**Coupled time $t$.** This component synchronizes the time steps used for student generation and discriminator evaluation. In the standard setup, the intermediate denoising time $\tau$ for the student's two-step generation ($\mathbf{z}_t \rightarrow \mathbf{z}_\tau$) and the corruption time $\tau'$ for preparing the discriminator's input ($\mathbf{x} \rightarrow \mathbf{z}_{\tau'}$ for corrupted student generations, and $\mathbf{x}' \rightarrow \mathbf{z}'_{\tau'}$ for corrupted teacher generations) are coupled, i.e., $\tau = \tau'$. The ablation ('w/o Coupled Time $t$') decouples them by sampling $\tau$ and $\tau'$ independently.

**Weight function** $\omega(t)$ **correction.** The gradient of the objective (5) is weighted by a time-dependent factor $\omega(t)$. The ablation ('w/o $\omega(t)$ Correction') uses a constant weight, i.e., $\omega(t) = 1$. Our full model applies a theoretically-motivated correction based on the noise schedule's signal rate $\alpha_t$, setting $\omega(t) \propto -\alpha'_t/(1 - \alpha_t)$, which re-weights the objective based on the difficulty of the denoising step at time $t$.

**Time scheduler** $\pi(\cdot)$ **weighting.** This controls how $\tau$ is sampled and (optionally) whether we apply importance weighting, which also refers to the sampling distribution $\pi(\tau)$ for the intermediate time step $\tau$. The ablation setting ('w/o $\pi(\tau)$ Weighting') samples $\tau$ from a uniform distribution. Our method employs a non-uniform Beta distribution as $\pi(\tau)$ to focus the student's training on specific, more informative intervals of the denoising trajectory. Specifically, we consider heavy-early (Beta$(2, 5)$), and heavy-late (Beta$(5, 2)$) schedules to improve sample quality. Practically, for *smaller NFEs* (e.g., 8 and 16 NFEs) we bias toward earlier times (e.g., Beta$(2, 5)$); for *larger NFEs* (e.g., 32, 64, and 128 NFEs) we bias toward later times (e.g., Beta$(5, 2)$).

**Regularization.** To stabilize training and maintain generalization, we introduce two regularization terms to the student's loss. The first is a KL divergence loss, which aligns the student's output logits with those of the frozen teacher at both steps of the generation process ($\mathbf{z}_t \rightarrow \mathbf{z}_\tau$ and $\mathbf{z}_\tau \rightarrow \mathbf{x}$, where $t = 1$ and $\tau \in (0, 1)$). Among Forward/Backward/Jeffreys implementations, we found forward KL consistently yielded the most stable optimization and best validation in our cases, so we adopt it for all ablations and apply it with a weight of $0.05$ in the student objective. The second term is an entropy bonus on the student's output distribution to encourage exploration, which is weighted by $0.0005$. The ablation ('w/o Regularization') removes both the KL-divergence and entropy terms from the student's objective.

**Guided inference.** As elaborated in Section 3.4, this technique is applied only during sampling to leverage the trained discriminator and improve sample quality. We employ a hybrid strategy that divides the denoising process into two phases. For the first 50% of NFEs, we use *gradient tilting* ($M = 1$), where the guidance scale $h$ is linearly increased from 30.0 to 40.0 to steer the generation. For the remaining 50% of NFEs, we switch to *multi-candidate re-ranking* ($h = 0$), sampling $M = 4$ candidates at each step and selecting the one with the highest reward. The ablation ('w/o Guided Inference') disables this process ($h = 0, M = 1$), reverting to standard unguided ancestral sampling.

## E.6 ABLATION STUDY RESULT ANALYSIS.

We now interpret the ablation results, breaking down our analysis by the function of each component group. We will discuss how the model is stabilized, how performance is driven by loss and time considerations, the conditional role of regularization, and finally, how inference-time guidance improves the final output.

**Baseline and score decomposition.** The baseline model (without tricks) performs poorly, with a Gen PPL of 803.9 at 8 NFEs. As shown in Table 1, incorporating only the Score Decompose provides a moderate initial improvement. However, its indispensable role is starkly revealed in our leave-one-out study (Table 2). Removing this component from the full model results in a catastrophic performance degradation (PPL$> 30,000$). This indicates a critical interplay: while a single-step gradient estimation is unstable within the complex training landscape created by our other optimizations, the two-step decomposition is essential for stabilizing the entire framework.

**Time coupling and loss weighting.** The most significant performance gain is achieved by introducing Coupled Time $t$, which dramatically reduces the 8-step PPL from 667.8 to 101.0. This highlights the importance of aligning the temporal context between the reward signal and the score function. Loss reweighting then refines convergence: the $\omega(t)$ Correction improves mid/late-step budgets (e.g., 32 NFEs: from 48.4 to 31.7; 64 NFEs: from 35.8 to 25.3), while $\pi(t)$ Weighting is especially effective at 16 NFEs ($75.6 \rightarrow 44.0$), with neutral to slight trade-offs elsewhere. The leave-one-out analysis confirms that removing any of these components leads to a severe degradation in performance.

**The dual role of regularization.** Our study reveals that KL/entropy regularization plays a dual role depending on the sampling budget: it is helpful at very small budgets, where discretization error might be severe. For very small NFEs (e.g., 8 NFEs), where large discretization errors can

destabilize training, regularization acts as a crucial stabilizer. The full model with regularization outperforms the version without it (PPL of 62.2 vs. 88.3). However, for more NFEs ($\geq 16$), Table 2 shows that **removing regularization yields superior results**, achieving the best PPLs in these settings (e.g., 30.99 vs. 38.19 at 16 NFEs). This suggests that for finer sampling schedules, the implicit stability of the chain is sufficient, and strong explicit regularization can over-constrain the model, leading to an accumulation of bias that harms the final generation quality.

**Guided inference.** Guidance is most effective at *small* NFEs, where it markedly lowers PPL. Specifically, at 8 NFEs, guidance reduces PPL from 88.3 to 62.2, a relative improvement of 29.6%. This trend continues at 16 NFEs, where PPL drops by 13.2% (from 44.0 to 38.2), and at 32 NFEs, where it decreases by 12.0% (from 28.4 to 25.0). Conversely, at high NFEs, the effect on PPL is negligible (e.g., from 21.95 to 21.91 at 64 NFEs). In this regime, however, guidance substantially improves sample diversity. We observe that entropy increases from 5.06 to 5.15 at 64 NFEs and from 5.00 to 5.15 at 128 NFEs, indicating that guidance can enhance variety without sacrificing quality. This pattern matches our hybrid schedule: early *gradient tilting* improves accuracy at small budgets, while late *multi-candidate re-ranking* expands support and boosts diversity at larger budgets.

### E.7 SENSITIVITY ANALYSIS OF RGAS

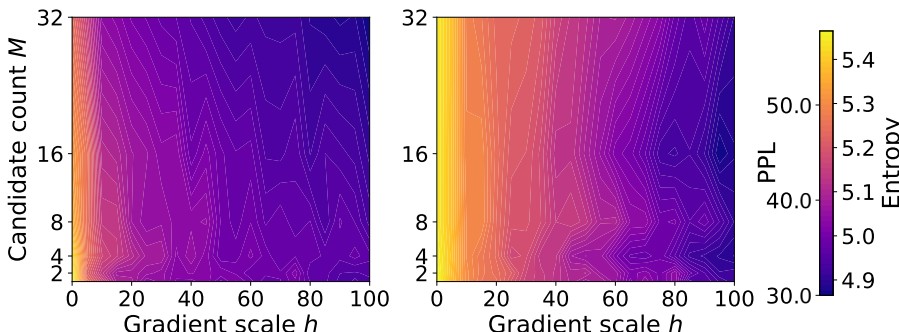

Figure 3: Sensitivity analysis of RGAS hyperparameters at 16 NFEs. We visualize the landscape of PPL (left) and Sequence Entropy (right) across varying gradient scales $h$, and candidate counts $M$. The heatmap indicates that increasing both $h$ and $M$ consistently reduces PPL, improving generation fidelity. However, larger gradient scales tend to lower sequence entropy, indicating a trade-off between sample quality and diversity.

To understand the interaction between the tilting scale $h$ and candidate number $M$ in RGAS, we conduct a comprehensive grid search to visualize the performance landscape, and a global sensitivity analysis using Functional ANOVA (fANOVA) (Hutter et al., 2014) to quantify hyperparameter importance.

**Performance landscape analysis.** We first study the joint effect of the RGAS hyperparameters $h$ and $M$ on generation quality. At a fixed budget of 16 NFEs and batch size 16, we perform a dense grid search, sweeping $h \in [0, 100]$ and $M \in \{1, 2, 4, 8, 16, 32\}$. Figure 3 presents the resulting PPL and Entropy. The resulting landscapes show a broad valley: increasing $h$ from 0 to 100 consistently reduces PPL, while entropy only gradually decreases and remains close to the teacher's entropy. This indicates that RGAS is relatively robust across a wide range of $(h, M)$ values and that there is no narrow "knife-edge" region required for good performance.

**Global sensitivity with fANOVA.** To formally quantify the relative importance of $h$ and $M$, we conduct a global sensitivity analysis using Functional ANOVA. We adopt a Latin Hypercube Sampling strategy and generate 1,000 hyperparameter pairs, with $h$ sampled continuously from $[20, 60]$ (a range that balances PPL and entropy) and $M$ sampled as an integer from $[1, 32]$. For each configuration, we run RGAS with 16 generated sequences and record the resulting PPL. This experiment is repeated separately for NFEs $= 8, 16, 32$, and the resulting performance is decomposed into variance contributions from $h$ and $M$.

Table 12 summarizes the fANOVA results. At 8 and 16 NFEs, $h$ explains around 90% of the variance in PPL, while $M$ accounts for less than 10%. At 32 NFEs, the importance of $M$ increases, but $h$

Table 12: Variance decomposition of PPL via fANOVA.

| Parameter | 8 NFEs | 16 NFEs | 32 NFEs |
|---|---|---|---|
| Gradient Scale ($h$) | 92.7% | 90.0% | 63.7% |
| Candidate Count ($M$) | 7.3% | 6.0% | 36.3% |

still contributes the majority of the variance. These findings suggest that: (i) in the low-NFE regime, $h$ is the dominant hyperparameter and should be prioritized during tuning; and (ii) as the sampling budget increases, $M$ becomes a more relevant knob for refining performance.

### E.8 MODEL SCALING UP

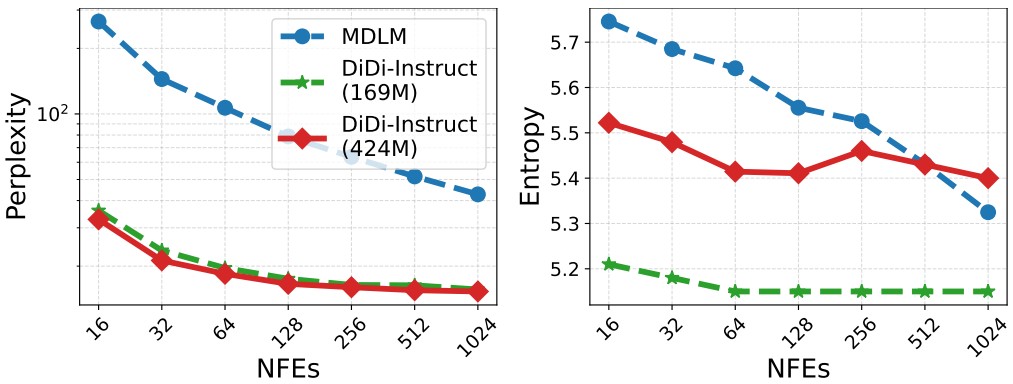

Figure 4: Scaling results for the 424M models. DiDi-Instruct significantly lowers PPL compared to the MDLM baseline across all NFEs.

To validate the effectiveness and scalability of DiDi-Instruct, we extended our experiments to a 424M parameter setting. In this evaluation, a pre-trained 424M MDLM served as the baseline model, where the teacher uses DiT+RoPE (more details can be found in Table 4) and is pre-trained for around 350,000 steps on 8×H100; the 424M student is then distilled on a single H100 in about 2 hours. We keep data, masking, optimizer, and schedule aligned with the 169M setting, and scale the discriminator to 373M with a deeper, SpecNorm+SiLU+Dropout head for per-token logits. We then applied DiDi-Instruct to produce the distilled student model. We assessed performance using two key metrics: PPL for predictive accuracy and token-level entropy for model confidence.

The results demonstrate a substantial improvement in language generation capabilities. As illustrated in Figure 4, our model achieves a significantly lower perplexity than the pre-trained baseline across all evaluated sequence lengths. The 424M distilled student model demonstrates remarkable efficiency, significantly outperforming the pre-trained base model's best-case performance (at 1024 NFEs). With only 16 NFEs, the student model already achieves a PPL that is $11.4\%$ lower than the base model at 1024 NFEs. This performance gap widens substantially with more inference steps: at 64 NFEs, the student's PPL is $56.8\%$ lower, and at 128 NFEs, it is $61.2\%$ lower. At the maximum of 1024 NFEs, the distilled model's PPL is $64.2\%$ lower than the base model's 1024-step result, showcasing a dramatic improvement in both generative quality and efficiency.

Entropy remains comparable while quality improves. The distilled 424M student tracks the base model closely across NFEs: the absolute gap is modest at low NFEs (e.g., 5.52 vs. 5.75 at 16 NFEs; $\Delta \approx -0.22$) and narrows as NFEs increase (5.43 vs. 5.43 at 512 NFEs; $\Delta \approx 0$), eventually turning slightly higher at 1024 NFEs (5.40 vs. 5.32; $\Delta \approx +0.08$). Overall, diversity is preserved while achieving markedly better PPL.

This incremental scaling experiment indicates that DiDi-Instruct quality–efficiency advantages persist at 424M with minimal procedural changes: large PPL reductions at matched NFEs and near-constant entropy. In summary, the 424M scale experiment confirms the robustness and scalability of DiDi-Instruct. Its ability to substantially improve upon an already capable, larger-scale base

model underscores its potential as an effective technique for distilling powerful and efficient student models.

## E.9 PROTEIN SEQUENCE GENERATION

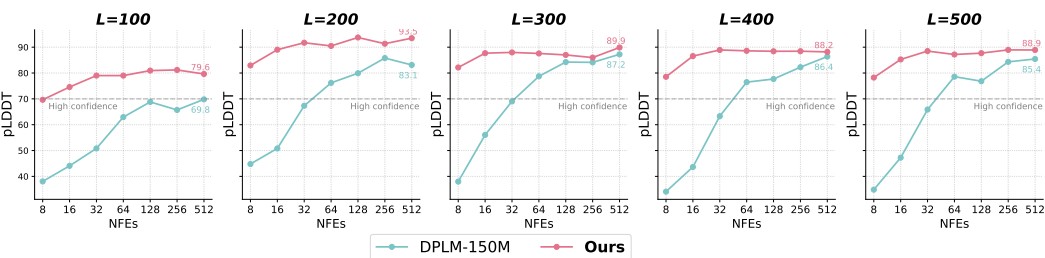

Figure 5: pLDDT comparison between DiDi-Instruct and the DPLM-150M across different sequence lengths ($L = 100, 200, 300, 400, 500$) and numbers of function evaluations (NFEs). Our method consistently outperforms the teacher, achieving up to +10 pLDDT gains at shorter sequence lengths (e.g., $L = 100$) and maintaining superior structural confidence across all lengths, even with substantially fewer sampling steps. Moreover, the distilled student exhibits more stable performance across NFEs, whereas the teacher shows larger variability as the number of steps increases.

**Dataset and evaluation metric.** The UniRef50 dataset (Suzek et al., 2014) contains approximately 45 million protein sequences, comprising roughly 14 billion amino acid tokens. Following previous work (Wang et al., 2024), we adopt a vocabulary of 33 tokens and a maximum sequence length of 1024, with longer proteins chunked into shorter chunks. We compute the predicted local distance difference test (pLDDT) score using the ESMFold model (Lin et al., 2023) to assess the foldability of the generated protein sequences, with values above 70 indicating high structural confidence. We report the pLDDT score across different sequence lengths and varying NFEs.

**Experiment details.** Our model architecture follows the DPLM-150M setting (Wang et al., 2024), with the student model and discriminator matched in size, while the discriminator incorporates a modified prediction head that outputs a scalar value. We set the learning rate of the student model to $1 \times 10^{-5}$ and that of the discriminator to $5 \times 10^{-6}$, with 1000 warm-up steps for each. All other training hyperparameters and sampling strategies follow those used in the text modeling task. The distillation of DPLM was completed in approximately two hours on a single H100 GPU, using batches of up to 4096 tokens.

**Protein sequence examples.** We provide visualizations of generated protein sequences to compare structural plausibility between the teacher and our distilled model. Figure 6 shows low-confidence outputs from the teacher under limited NFEs, while Figure 7 presents high-confidence examples produced by DiDi-Instruct in Figure 7.

**Sequence diversity analysis.** High predicted structural confidence (pLDDT) can sometimes indicate a risk of mode collapse, where the model generates repetitive samples. Therefore, we evaluate sequence-level diversity using MMseqs2 clustering (Steinegger & Söding, 2017) with a 30% sequence identity and 80% coverage threshold—standard criteria for determining biologically meaningful variation. Across low-NFE settings (32–64), DiDi-Instruct exhibits diversity comparable to DPLM, which reflects the stochastic nature of early denoising. As the number of sampling steps increases (128–512 NFEs), DiDi-Instruct yields slightly lower entropy and moderately larger cluster sizes, indicating a more concentrated output distribution. Importantly, diversity levels remain competitive with DPLM across all settings, and no regime shows abnormal cluster growth or a collapse of entropy. Noted that genuine mode collapse in this setting would correspond to near-zero cluster entropy together with an average cluster size approaching the full sample count, i.e., all sequences falling into a single MMseqs2 cluster. This collapse behavior is not observed in any configuration.

This behavior is consistent with our text-generation ablations, where diversity naturally decreases as NFEs grow and the denoising trajectory becomes more deterministic. Moreover, DiDi-Instruct is explicitly designed for *few-step* generation; mild diversity reduction at very large NFEs does not contradict our intended operating regime. Overall, these results confirm that DiDi-Instruct maintains

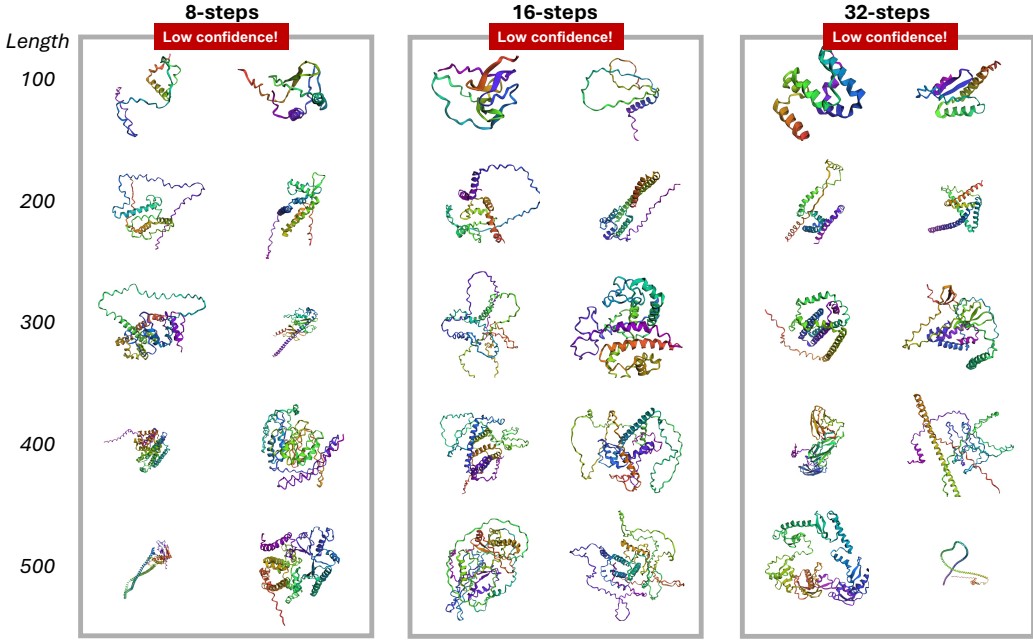

Figure 6: Visualization examples of low-confidence protein sequences (pLDDT < 70) generated by the teacher model with lengths ranging from 50 to 500, under 8, 16, and 32 NFEs

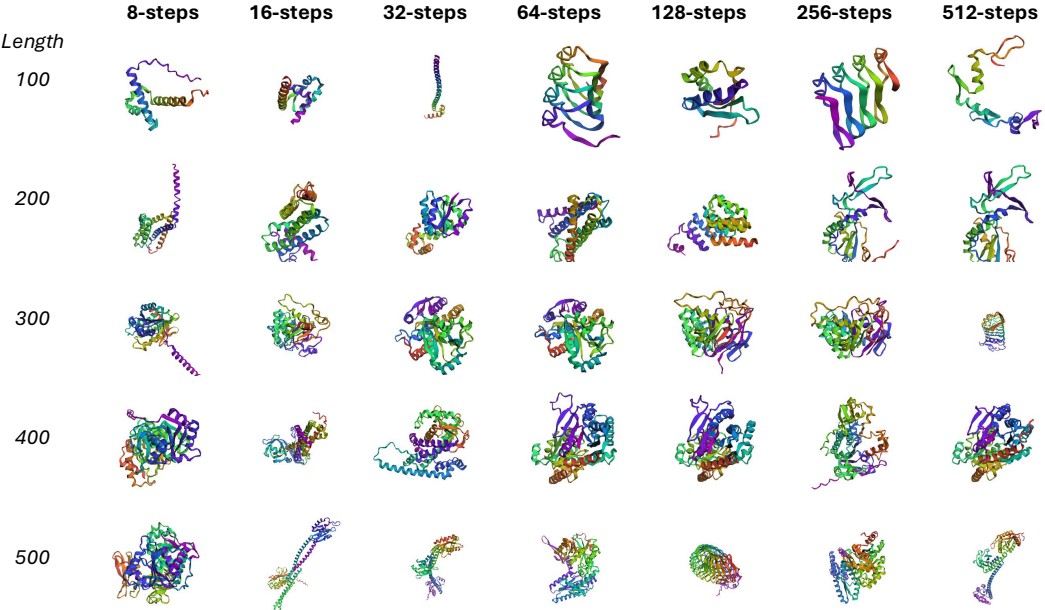

Figure 7: Visualization examples of high-confidence protein sequences (pLDDT > 70) generated by the DiDi-Instruct model, with lengths ranging from 50 to 500 across different numbers of function evaluations (NFEs).

biologically meaningful variability across the generation budget while avoiding collapse to a small subset of high-pLDDT sequences.

Table 13: Protein sequence diversity evaluation using MMseqs2 clustering. We compare the Average Cluster Size and Cluster Entropy across different NFEs.

| NFEs | Methods | Cluster Entropy ↑ | | | | | Average Cluster Size ↓ |
|---|---|---|---|---|---|---|---|
| | | 100 | 200 | 300 | 400 | 500 | |
| 32 | DPLM | 3.68 | 3.68 | 3.55 | 3.65 | 3.68 | 1.02 |
| | **DiDi-Instruct (Ours )** | 3.68 | 3.52 | 3.15 | 3.58 | 3.55 | 1.11 |
| 64 | DPLM | 3.68 | 3.51 | 3.45 | 3.68 | 3.53 | 1.07 |
| | **DiDi-Instruct (Ours)** | 3.68 | 3.65 | 3.45 | 3.45 | 3.54 | 1.09 |
| 128 | DPLM | 3.68 | 3.65 | 3.50 | 3.52 | 3.61 | 1.05 |
| | **DiDi-Instruct (Ours)** | 3.62 | 3.20 | 3.42 | 3.45 | 3.38 | 1.21 |
| 256 | DPLM | 3.61 | 3.42 | 3.41 | 3.19 | 3.44 | 1.19 |
| | **DiDi-Instruct (Ours)** | 3.61 | 3.11 | 3.41 | 2.87 | 3.07 | 1.40 |
| 512 | DPLM | 3.58 | 3.23 | 3.02 | 3.11 | 2.97 | 1.41 |
| | **DiDi-Instruct (Ours)** | 3.54 | 3.11 | 2.95 | 2.77 | 2.79 | 1.53 |

### E.10 TEXT EXAMPLES

To provide a more intuitive understanding of the performance of our distillation framework, this section contains qualitative examples of generated text. We present a side-by-side comparison of samples generated by the original 1024-step pretrained teacher model and our distilled DiDi-Instruct student. For the student, we showcase generations from a range of few-step sampling configurations: $8, 16, 32, 64$, and 128 NFEs. This allows for a direct visual inspection of the trade-offs between computational cost (i.e., NFEs) and sample quality.

We here provide a qualitative analysis of generated texts to evaluate our models across several key linguistic dimensions. The results indicate that as the number of function evaluations (NFEs) increases, DiDi-Instruct students show marked improvements in coherence and narrative consistency, eventually outperforming the teacher. Specifically, we evaluate the generated texts with respect to the following three criteria:

**Fluency and repetition.** All models (including the teacher and students) generally produce fluent and grammatically correct sentences. The most significant exception is the 8-step student, which suffers from severe phrasal repetition (e.g., repeating "affirmative action program"), a common artifact of extreme model compression. This issue is effectively eliminated in student models with 16 NFEs or more, which exhibit fluency comparable to the teacher.

**Coherence and topic adherence.** This is where the most critical differences emerge. The 1024-step teacher struggles with global coherence, frequently shifting between unrelated topics within a single output. In contrast, while the 8-step student is incoherent due to repetition, the 16-step student already demonstrates stronger paragraph-level topic adherence than the teacher. This ability to maintain a consistent narrative thread strengthens progressively. The 128-step student excels at this, developing a central theme with supporting details over a longer text, showcasing high global coherence.

**Informativeness and specificity.** The informativeness of the generated text correlates strongly with coherence. The teacher's outputs, despite containing specific details, are uninformative as a whole due to topic shifts. The 8-step student's text has very low information content due to repetition. As NFEs increase from 16 to 128, the students' generations become increasingly specific and informative. For example, the 64-step model constructs a detailed news-style report, and the 128-step model builds a multi-faceted story, both rich in contextual details.

Overall, the generated text samples show that DiDi-Instruct successfully distills the teacher's linguistic capabilities while simultaneously improving its ability to construct coherent and focused narratives.

---

**Text obtained with 1024 NFEs teacher model. Perplexity=40.48, Entropy=5.22.**

```
He was mid-morning early, and I started forming a motorcade for him. He
 ↪  stopped every NYPD car, and I radioed back to every bicycle
 ↪ officer that was in there, to protect him. I did not hear you said
 ↪ to be involved in it.

That you were involved with that campaign. I never asked anybody to
 ↪ test it. You did need help you leaked it, not made it for yourself.
 ↪  He parked and it was then that I saw that president and ended up
 ↪ there, speaking with a journalist.

The Prince was granted anonymity and noted that the FBI was aware of
 ↪ the familiarity with Russia and was conducting the investigation as
 ↪  appropriate. In some accounts, Casey Wilson, senior counsel for
 ↪ the Bonden Bisson, told The Post that the President is aware of the
 ↪  inquiry.

But Mr. Baker's testimony is the first time someone has long taken
 ↪ something of concern into the hands of their officials probing
 ↪ Trump's personal relationships with foreign officials. Former FBI
 ↪ regional director James Kogan, a former officer who met with
 ↪ President Trump, was once briefed about the inquiry from the
 ↪ Russian District Attorney General.

Victor Atkinson, an FBI chief-officer who is retired, hacked the
 ↪ personal PINs and cell phone lines of the two Trump officials to
 ↪ conceal the information, as shown inside the country's electronic
 ↪ networks. Anonymous messages can appear in our apps like Twitter
 ↪ and Facebook and if one's political actions resists they become
 ↪ blocked.
```

---

**Text obtained with 8 NFEs distilled by DiDi-Instruct. Perplexity=62.24, Entropy=5.17.**

```
An engineer's run up against a system of dynamic memory-producing
 ↪ technology. An examiner's mechanism of memory has to reduce
 ↪ property crimes against the technology.

India's memory system has run into a deficit of memory and that
 ↪ economic damage in India has run up against a system of Now and
 ↪ Then. The new system will be designed as an economically secure and
 ↪  efficient system in determining how residents receive support.

A loss of support from memory management systems won't go too far from
 ↪ promise. With a deficit of 1 million for residents, the Navajo
 ↪ Nation is testing a new system that won't prevent you from
 ↪ receiving your financial stream.

The Navajo Nation faces a deficit of 1.88 million in systems that are
 ↪ designed to enhance what residents receive. It's a potential loss
 ↪ of support from a system that has been receiving both positive and
 ↪ negative feedback from the community.

A presidential candidate who once served during four tours in Iraq and
 ↪ Afghanistan refers to his new record as a figurative use of pop
 ↪ culture. I got a disappointed order from the State Department, the
 ↪ new record is to go against the American health care system.
```

Text obtained with 16 NFEs distilled by DiDi-Instruct. Perplexity=30.99, Entropy=5.22.

```
by a biomedical engineer and Justin Zetter, a researcher at Case
  ↪ Western University who developed a non-profit. Zetter says that the
  ↪  sperm the new lab produces is quickly going to a place better
  ↪ stores and than human sperm.

A year and a half years ago Zetter took out of this lab and coated it
  ↪ with a chicken, and he is optimistic it will come in next year. "It
  ↪ 's very exciting to see animal entrepreneurship in the future," he
  ↪ says.

Foreign Minister Chid wants to use a Frieses system for links to Saudi
  ↪ Arabia. The Saudi interior minister has announced a plan to allow
  ↪ imports to use a Frieses system system that links the Gulf nation's
  ↪  ferries.

While the government had initially rejected the system, the Saudi
  ↪ Statesport Service post has announced that it is considering
  ↪ amalgamating two ferries including ElKhasi Oera Tries, which was
  ↪ run-owned by Sad Oera in Qatar, to avoid links to Riyadh.

Mohsen Chid, a spokesman for Saudi Arabia's Public Security Ministry,
  ↪ said Monday that the ministry is considering a plan to use a
  ↪ Frieses and trip management system known as ElKhasi Oera Tries,
  ↪ which was founded in 1927.
```

Text obtained with 32 NFEs distilled by DiDi-Instruct. Perplexity=23.60, Entropy=5.20.

```
A Virginia man accused of being involved in a car crash last month
  ↪ initially thought a first-year student in therapy during a blocked
  ↪ light video, according to a North Carolina grand jury. Khalid
  ↪ Anthony Matteo told investigators he was spending night in therapy.

Matteo was arrested after being prescribed a Virginia painkiller for
  ↪ blocking someone's vehicle during a display of headlights. Federal
  ↪ prosecutors have recommended nearly 30 months of outpatient
  ↪ treatment for Matteo, who was in the first year of an outpatient
  ↪ treatment program.

Prosecutors initially said a woman had spent the night in therapy to
  ↪ avoid the video, but Matteo was given a month of outpatient
  ↪ treatment and is up to nearly 30 years of probation. A Democratic
  ↪ nominee for the presidential nomination sought the support of state
  ↪  lawmakers.

The nominee won't block any state cuts that they promise otherwise. Del
  ↪ . Tom Perez will give testimony before the bench of the District
  ↪ Court, after a group of Virginia lawmakers would cut short the
  ↪ state's responsibility for the federal health care plan.

The plan was designed to ensure that seniors receive access to health
  ↪ care. Virginia lawmakers say the federal plan would allow all
  ↪ seniors to have access to health care, but lawmakers have
  ↪ threatened to block its implementation over claims that it would
  ↪ reduce the deficit.
```

Text obtained with 64 NFEs distilled by DiDi-Instruct. Perplexity=19.61, Entropy=5.18.

```
People gathered at the scene of an autopsy of a dead Egyptian army
  ↪ soldier near the northern Sinai town of Khaled el-Zawiya, east of
  ↪ the Egyptian capital, Cairo, November 9, 2017. REUTERS/Stringer
  ↪ Sundays name was not immediately released.

A police official in Cairo, Mohamed Hussein, said that the soldier was
  ↪ involved in a confrontation with a group of protesters acting
  ↪ counter-duty and fired projectiles at him, Reuters said. The U.N.
  ↪ investigators said that Egyptian forces fired tear gas.

When hundreds of protesters gathered in the desert near Sheikh el-
  ↪ Zawiya, near the hometown of the late Egyptian Abdel Fattah el-Sisi
  ↪ . The UN initially accused the Egyptian forces of using excessive
  ↪ force during the confrontation, but later apologized.

Kharyl Hershehey claims he was driving night-in his Mercedes-Benz Audi
  ↪ over night to avoid a trip. An Ohio court-appointed attorney is a
  ↪ Virginia man accused for drinking someone while he tried to drive
  ↪ his green car over parked near a stop.

The trip was leading to a risk of a head injury. Kharyl Joshua
  ↪ Hershehey is accused of drinking someone while driving his car in a
  ↪  real green engine to avoid a trip, say court documents. The
  ↪ drunken driver claims a woman was killed.
```

Text obtained with 128 NFEs distilled by DiDi-Instruct. Perplexity=17.50, Entropy=5.17.

```
Officials at the U.S. District of Virginia claim a "fatality epidemic"
  ↪ is causing deaths in school classrooms around the U.S. The Virginia
  ↪  school district is investigating the death of student Tamey Nelson
  ↪  after she wielded a knife to kill a classmate in class.

A school in Virginia is investigating after state claims that a "
  ↪ fatality epidemic death" is causing deaths among students in
  ↪ classrooms around the world. Officials at the U.S. district
  ↪ disputed the school's claims of an outbreak.

Two students were killed and three others were hospitalized, according
  ↪ to a news report obtained by NBC News. "I am concerned about the
  ↪ safety of children and students in classrooms in the United States
  ↪ ," the District of Virginia District Attorney James Trotsma told
  ↪ NBC News.

Parents of a Virginia school claim that a "child's life of death" was
  ↪ caused by an epidemic of abuse in classrooms around the U.S.
  ↪ district on Sunday, according to a news report. A video surfaced of
  ↪  a 12-year-old student being struck and killed.

The student was killed by a knife wielded by others they identified as
  ↪ "revered students," a similar incident described by a teacher that
  ↪ was caused by the same incident in class at the school's Iron
  ↪ Arlington Branch Learning Center.
```

