# OpenReview forum: "Ultra-Fast Language Generation via Discrete Diffusion Divergence Instruct"
_ICLR.cc/2026/Conference — ICLR 2026 Poster_

### Official Review · Reviewer_n4tg · 2025-10-27

**Soundness:** 3
**Presentation:** 3
**Contribution:** 3
**Rating:** 6
**Confidence:** 2

**Summary:**

The paper proposes Discrete Diffusion Divergence Instruct (DiDi-Instruct), a distillation framework that trains a few-step discrete diffusion language model (student) to match a pre-trained masked diffusion language model (teacher) by minimizing an integral KL divergence (IKL) across noise levels. The core gradient is rewritten as a score-function estimator weighted by a log density-ratio reward between student and teacher; the reward is estimated via an auxiliary discriminator. Training is stabilized by grouped reward normalization and a score decomposition that exposes the student to intermediate corruption states. At inference, a Reward-Guided Ancestral Sampler (RGAS) uses discriminator signals for gradient tilting in early steps and re-ranking in later steps. On OpenWebText, the model achieves perplexity 62.2- 18.4 at 8 - 128 NFEs, with small entropy loss and lower distillation cost, and it scales to larger models and a protein-sequence task.

**Strengths:**

1. The paper adapts IKL-based distribution matching to masked discrete diffusion and derives a policy-gradient form that bypasses non-differentiable paths, with a discriminator estimating the log density ratio used as reward. This method is different from time-matching distillations and provides a route to few-step discrete generation.
2. The derivation gives the IKL objective and yields a score-function identity in which the reward is the marginal log-density ratio between student and teacher; the proof sketch and assumptions are stated, and the link to a tractable discriminator estimator is explicit.
3. On OpenWebText the method reports consistently lower PPL across 8–128 steps, with faster distillation and small entropy loss. The paper includes cumulative and leave-one-out ablations indicating gains from the proposed components, and shows transfer to protein sequence generation.
4. The paper is clearly written and well structured.

**Weaknesses:**

1. The comparison to the baselines are is too abbreviated with only one figure of perplexity. It would be better to have a table presenting more metrics besides the ppl.
2. DiMO is discussed in the related work in appendix but is not include as an experiment baseline.

**Questions:**

1. Can you report tokens/sec (or sequences/sec) and latency per 1k tokens at matched perplexity versus a similarly sized AR model with and without speculative decoding, on the same hardware and kv-cache settings? This would clarify the practical speed–quality trade-off beyond NFEs.
2. Can you include downstream tasks? A single perplexity metric is not enough to establish practical advantage of the new method.

---

> ### Author Response · Authors · 2025-11-21
> **Response to Reviewer n4tg [1/2]**
>
> We highly appreciate your time and the insightful feedback. Changes are marked in **green** (for your comments) and **purple** (for comments from multiple reviewers) in our revision.
>
> > **Q1.** It would be better to have a table presenting more metrics besides the PPL.
>
> **A1.** Thanks for this insightful suggestion. To provide a comprehensive comparison, we consider *MAUVE* (for distribution similarity) and *Self-BLEU* (for diversity). All models generate 1,024 unconditional samples with identical decoding configurations (no prompt, max length 1024). All baselines use the same sampling budgets (8/16/32/64/128 NFEs), and all samples are tokenized and detokenized using the same GPT-2 tokenizer.
>
> **Self-BLEU** is computed with n=5, uniform n-gram weights, and smoothing method 1. We tokenize each sample using NLTK’s word_tokenize and compute the average BLEU score by treating each sample as the hypothesis and all other samples as references.
>
> **MAUVE** is computed using the official ‘mauve-text’ implementation, which internally uses GPT-2–based embeddings. All generated and reference samples are evaluated with a maximum text length of 256 tokens, as recommended by the library default.
>
> | Metrics              | Methods       | 8     | 16    | 32    | 64    | 128   |
> |------------|-------------|-------|-------|-------|-------|-------|
> | MAUVE $\uparrow$ ($\times 10^{-3}$)     | MDLM          | 4.77  | 6.76  | 6.46  | 5.50  | 5.97  |
> |                      | SDTT          | 5.60  | 5.38  | 5.70  | 5.04  | 5.96  |
> |                      | DUO           | 5.75  | 5.15  | 5.99  | 5.85  | 6.05  |
> |                      | DiDi-Instruct | 5.84  | 6.55  | 6.54  | 6.14  | 6.31  |
> | Self-BLEU $\downarrow$ ($\times 10^{-2}$) | MDLM          | 2.91  | 4.58  | 6.36  | 7.97  | 9.33  |
> |                      | SDTT          | 5.02  | 7.98  | 10.09 | 12.49 | 12.92 |
> |                      | DUO           | 8.26  | 9.28  | 9.62  | 8.84  | 8.94  |
> |                      | DiDi-Instruct | 4.82  | 4.64  | 5.24  | 6.25  | 8.24  |
>
> **Conclusions**: Our method improves both diversity and fidelity through *score decomposition* (Eq. 9) and an *adversarial distribution-matching objective*. Regression-based baselines such as SDTT tend to prioritize pointwise accuracy, which inadvertently drives the model toward mode-seeking behavior and worse Self-BLEU. In contrast, score decomposition forces the student to model the evolution of noisy states, thereby preserving diverse modes. Simultaneously, adversarial training ensures that the learned marginal distribution matches that of the teacher, which produces samples with higher distributional fidelity as evidenced by improved MAUVE.
>
> > **Q2.** It would be good to clarify the practical latency (tokens/sec) and speed-quality trade-off.
>
> **A2.** We appreciate this constructive suggestion. In the rebuttal phase, we ran a practical latency benchmark and found DiDi-Instruct achieves around **13.2x reduction in latency** (0.423s vs 5.719s per 1k tokens) while matching the AR model's PPL.
>
> - Setup: We consider 1x H100, bfloat16 precision, a batch size of 16, and 1024-length sequences. All models are 169M (12 layers, 768 hidden, 12 heads). The AR baseline employs greedy decoding and benefits from standard optimizations (e.g., KV-caching, FlashAttention).
>
> - Note that this 13.2x gain is a conservative estimate without infrastructure-level optimizations. DiDi-Instruct is fully compatible with hybrid AR–diffusion designs [1] and KV-cache reuse + parallel decoding [2], which provide substantial additional possibilities for accelerating inference.
>
> | Models          | NFEs   | Tokens/sec $\uparrow$  | Latency per 1k tokens $\downarrow$ |
> | ----------------- | ------ | ------------ | --------------------- |
> | ARM                | 1024   | 179.042      | 5.719 s               |
> | MDLM             | 512    | 111.146      | 9.213 s               |
> | SDTT              | 64     | 806.498      | 1.269 s               |
> | **DiDi-Instruct** | **16** | **2366.604** | **0.432 s**           |
>
> **Conclusion**: The results demonstrate that DiDi-Instruct achieves a substantially better speed–quality trade-off than existing baselines, which delivers high-quality generation at an order-of-magnitude lower latency.
>
> **References:**
>
> [1] Block Diffusion: Interpolating Between Autoregressive and Diffusion Language Models. ICLR 2025
>
> [2] Fast-dLLM: Training-free Acceleration of Diffusion LLM by Enabling KV Cache and Parallel Decoding. arXiv:2505.22618. 2025

---

> ### Author Response · Authors · 2025-11-21
> **Response to Reviewer n4tg [2/2]**
>
> > **Q3.** It would be great to include DiMO as an experiment baseline.
>
> **A3.** Sorry for the confusion. We will add a section in the Appendix to discuss DiMO and the modality gap behind them.
>
> In practice, A direct extension of DiMO to our setting yields comparable generative PPL but significantly degraded diversity (entropy around 4.0 vs. 5.22 for DiDi-Instruct). This behavior suggests training instability and potential mode collapse under a naive adaptation.
>
> We think this phenomenon comes from the significant **modality gap**. To the best of our knowledge, DiMO is specialized for text-to-image generation, where the diffusion process operates on image codebook tokens characterized by *2D spatial dependencies* and visual coherence. This differs fundamentally from pure language generation, which necessitates modeling *1D sequential dependencies* and strict syntactic constraints. Consequently, transferring the architectural choices tuned for visual latents to the text domain is non-trivial and likely explains the suboptimal behavior in our experiments. We greatly appreciate your suggestion, and we consider a principled, text-specific adaptation of DiMO to be a valuable area for future work.
>
> > **Q4.** It would be great for authors to include downstream tasks.
>
> **A4.** We further include downstream evaluations covering **(1) Domain Adaptation** (two tasks, MMLU and PubMed) and **(2) Frozen Feature Extractor Evaluation** to address your concern.
>
> **(1) Domain Adaptation**
>
> **a) MMLU Supervised Fine-Tuning.**
> We fine-tune the teacher MDLM, SDTT, and DiDi-Instruct (all 169M parameters) on the MMLU benchmark [3]. All models are trained for 5,000 SFT steps, and we report validation NLL and accuracy. DiDi-Instruct matches the teacher’s downstream performance and achieves the largest NLL improvement:
>
> | Model             | NLL (before) | NLL (after) $\downarrow$ | ΔNLL $\uparrow$    | ACC (before) | ACC (after) $\uparrow$ | ΔACC $\uparrow$ |
> | ----------------- | ------------ | ------------- | --------- | ------------ | ------------- | ------ |
> | MDLM              | 9.879        | 2.814         | 7.065     | 20.6%        | 25.3%         | 4.7%   |
> | SDTT              | 9.773        | 2.844         | 6.929     | 21.2%        | 24.3%         | 3.1%   |
> | **DiDi-Instruct** | 9.960        | 2.815         | **7.145** | 21.6%        | 25.1%         | 3.5%   |
>
> **b) Scientific Domain Adaptation (PubMed).**
>
> We further evaluate domain transfer by performing a short 5k-step LM-objective SFT on the PubMed corpus. All models improve significantly, and DiDi-Instruct nearly matches the teacher (24.944 → 24.894 PPL) while outperforming SDTT and DUO:
>
> | Model             | NLL (before) | PPL (before) | NLL (after) $\downarrow$ | PPL (after) $\downarrow$ |
> | ----------------- | ------------ | ------------ | ----------- | ------------- |
> | MDLM              | 3.761        | 42.983       | 3.217       | 24.944        |
> | SDTT              | 3.938        | 51.336       | 3.243       | 25.602        |
> | DUO               | 4.028        | 56.174       | 3.338       | 27.734        |
> | **DiDi-Instruct** | 3.879        | 47.563       | **3.215**   | **24.894**    |
>
> These results confirm that **DiDi-Instruct remains highly adaptable to new scientific domains** while maintaining its efficiency gains.
>
> **(2) Feature Extractor Evaluation**
>
> We also evaluate the models in a **frozen-feature setting** using the GLUE MRPC benchmark [4]. The pretrained backbone (MDLM / SDTT / DiDi-Instruct) is fully frozen, and only a small classification head is trained. This isolates the quality of pretrained semantic representations.
>
> | Model             | ACC $\uparrow$     | F1 $\uparrow$      |
> | ----------------- | ---------- | ---------- |
> | MDLM             | 59.6%     | 72.2%     |
> | SDTT              | 59.1%     | 72.0%     |
> | **DiDi-Instruct** | **62.3%** | **73.3%** |
>
> **Takeaway**: DiDi-Instruct achieves the best accuracy and F1 among all models, demonstrating that DiDi-Instruct does not degrade representation quality and can even slightly improve semantic discrimination compared to the MDLM teacher. This complements our generative results and shows that **DiDi-Instruct retains strong downstream utility under both fine-tuning and frozen-encoder conditions**.
>
>
> **References**
>
> [3] Automatically Constructing a Corpus of Sentential Paraphrases. IWP 2005
>
> [4] Measuring Massive Multitask Language Understanding. ICLR 2021
>
> **Dear reviewer, we hope our rebuttal has fully addressed all of your concerns. If you still have remaining concerns, please do let us know. We are very glad to bring more clarifications!**
>
> Best regards,  authors of the submission #19443

---

### Official Review · Reviewer_HAtL · 2025-10-30

**Soundness:** 2
**Presentation:** 3
**Contribution:** 3
**Rating:** 6
**Confidence:** 4

**Summary:**

This paper introduces a training-based distillation framework DiDi-Instruct to accelerate generation for discrete diffusion language models (dLLMs). The core idea of DiDi-Instruct is to distill a pre-trained teacher dLLM into a student model that can generate high-quality text in very few steps. The method is centered on minimizing the Integral KL (IKL) divergence between the teacher and student models, which forces the student to match the teacher's data distribution. The author proposes comprehensive solutions to make the training on discrete data tractable, concluding: 1) inspired by policy gradient, decomposing the gradient of the objective as a score function weighted by a log-density ratio to avoid non-differentiable issue. 2) using an auxiliary discriminator to approximate the log-density ratio because it is intractable to compute directly. 3) leveraging grouped reward normalization to stabilize reward. The training is end-to-end and jointly for discriminator and student model. During inference, the author uses the trained discriminator to guide the sampling process and improve output quality. Experiments show the effectiveness of the proposed method that DiDi-Instruct significantly outperforms existing accelerated models and the GPT-2 baseline in perplexity, and the distilled model surpasses the 1024-step teacher's quality with only 16 steps.

**Strengths:**

1. The paper presents a mathematically rigorous solution to the difficult problem of distillation in a discrete state space. Instead of relying on heuristics, it adapts the IKL divergence objective from continuous models and  reformulates it as a tractable policy gradient problem.
2. The method consistently outperforms other distillation methods and the GPT-2 baseline in perplexity across all sampling budgets.
3. The proposed method demonstrates high efficiency. The training only needs one GPU, compared with 8 GPU for training teacher model. For generation, it is able to surpass the text quality of its 1024-step teacher model with only 16 inference steps, representing a significant acceleration.
4. The proposed method avoids the mode collapse problem, which is a common problem in distillation. Experimental results on text generation demonstrate negligible entropy loss.

**Weaknesses:**

1. The author performs distillation on a self-trained teacher model. This raises questions about the method's transferability and effectiveness when applied to other pre-trained, open-source dLLMs, such as LLaDA [1] or Dream [2].
2. In the text generation experiments, the evaluation is limited to perplexity and entropy. These metrics alone cannot accurately reflect the true text generation capabilities. The authors should conduct a more comprehensive evaluation on standard text generation benchmarks (e.g., MMLU, GSM8K, Humaneval) to demonstrate that the student model can indeed achieve performance comparable to the teacher model while using fewer steps.
3. In the protein sequence experiments, the pLDDT score increases significantly after distillation. However, excessively high pLDDT scores might indicate a potential risk of mode collapse (e.g., collapsing to a few sequences with the highest pLDDT, thereby failing to sample other valid sequences), which will lead to a reduction in generation diversity. The authors need to clarify the generation results from the perspective of protein sequence diversity (e.g., by using MMseqs clustering) to address this concern.

[1] Dream 7B: Diffusion Large Language Models

[2] Large Language Diffusion Models

**Questions:**

See above weaknesses.

---

> ### Author Response · Authors · 2025-11-21
> **Response to Reviewer HAtL [1/2]**
>
> We appreciate the helpful feedback. We put all revisions in the manuscript colored in **blue** (related to your specific concerns) and **purple** (to common comments from multiple reviewers). In the following paragraphs, we will address your concerns one by one.
>
> > **Q1.** Distill Open-source dLLMs such as LLaDA or Dream
>
> **A1.** We appreciate your sharp intuition and agree that validating scalability in LLaDA and Dream is crucial and promising. We would like to highlight a very recent work [1] (Nov 2025) that adopts a similar $\textcolor{green}{\text{adversarial-distillation methodology}}$  for $\textcolor{green}{\text{GPT-5 distillation}}$. Their key techniques for achieving  $\textcolor{green}{\text{stable and scalable}}$ training include:
>
> - Using GPT-5 as the teacher and a fine-tuned **Qwen-2.5-14B model** as the student (theirs is in smaller models in LLMs; $\textcolor{green}{\text{ours in few-step/fast generations of diffusion LLMs}}$).
>
> - On-policy training using GRPO ($\textcolor{green}{\text{same as ours}}$).
>
> - Employing an adversarial-distillation loss as the primary training objective  ($\textcolor{green}{\text{conceptually aligned with ours}}$).
>
> - Introducing a warm-up phase for the discriminator to stabilize the early stages of training ($\textcolor{green}{\text{same as ours}}$).
>
> This also motivates and reinforces our direction in distilling LLaDA & Dream in future research: by reducing generation steps, we can empower large-scale dLLMs to achieve fast inference speeds. We are actively conducting these experiments to demonstrate this unique efficiency advantage at scale.
>
> > **Q2.** Consider MMseqs clustering for protein sequence diversity evaluation
>
> **A2.** To address your concern, we evaluate sequence-level diversity using **MMseqs2 clustering** [2] under a 30% identity and 80% coverage threshold, a standard criterion for measuring biologically meaningful variation. For each model and diffusion step budget, we generate *40 sequences* across five protein lengths (100–500) and compute both the *cluster entropy* and the *average cluster size*.
>
> | Method (32 NFEs) | 100  | 200  | 300  | 400  | 500  | avg cluster size |
> | -- | -- | -- | -- | -- | -- | -- |
> | DPLM              | 3.68 | 3.68 | 3.55 | 3.65 | 3.68 | 1.02             |
> | **DiDi-Instruct**          | 3.68 | 3.52 | 3.15 | 3.58 | 3.55 | 1.11             |
>
> | Method (64 NFEs) | 100  | 200  | 300  | 400  | 500  | avg cluster size |
> | -- | -- | -- | -- | -- | -- | -- |
> | DPLM              | 3.68 | 3.51 | 3.45 | 3.68 | 3.53 | 1.07             |
> | **DiDi-Instruct**          | 3.68 | 3.65 | 3.45 | 3.45 | 3.54 | 1.09             |
>
> | Method (128 NFEs) | 100  | 200  | 300  | 400  | 500  | avg cluster size |
> | -- | -- | -- | -- | -- | -- | -- |
> | DPLM               | 3.68 | 3.65 | 3.50 | 3.52 | 3.61 | 1.05             |
> | **DiDi-Instruct**           | 3.62 | 3.20 | 3.42 | 3.45 | 3.38 | 1.21             |
>
> | Method (256 NFEs) | 100  | 200  | 300  | 400  | 500  | avg cluster size |
> | -- | -- | -- | -- | -- | -- | -- |
> | DPLM               | 3.61 | 3.42 | 3.41 | 3.19 | 3.44 | 1.19             |
> | **DiDi-Instruct**           | 3.61 | 3.11 | 3.41 | 2.87 | 3.07 | 1.40             |
>
> | Method (512 NFEs) | 100  | 200  | 300  | 400  | 500  | avg cluster size |
> | -- | -- | -- | -- | -- | -- | -- |
> | DPLM               | 3.58 | 3.23 | 3.02 | 3.11 | 2.97 | 1.41             |
> | **DiDi-Instruct**           | 3.54 | 3.11 | 2.95 | 2.77 | 2.79 | 1.53             |
>
> **Analysis:** Across low-NFE settings (32–64), both DiDi-Instruct and DPLM produce similarly high sequence diversity under MMseqs2 clustering, which reflects the stochastic nature of early denoising. As NFEs increase (128–512), DiDi-Instruct shows a moderately more concentrated output distribution, manifested as slightly lower cluster entropy and somewhat larger average cluster sizes. This trend is expected: distillation encourages the student to follow a shorter and more deterministic denoising trajectory as the sampling budget grows. Crucially, the diversity levels remain comparable to DPLM across all budgets, and we **do not observe the characteristic signatures of mode collapse** (e.g., entropy approaching zero or almost all samples falling into a single cluster).
>
> **Reference**
>
> [1] Black-Box On-Policy Distillation of Large Language Models. arXiv:2511.10643. Nov 13 2025
>
> [2] MMseqs2 Enables Sensitive Protein Sequence Searching for the Analysis of Massive Data Sets. Nature Biotechnology. 2017.

---

> ### Author Response · Authors · 2025-11-21
> **Response to Reviewer HAtL [2/2]**
>
> > **Q3.** Consider standard text generation benchmarks
>
> **A3.** We appreciate your insightful suggestion. Motivated by your suggestion and also related ones from other reviewers, we conducted new experiments to compare models in different ways: **(1) domain adaptation** (two tasks, MMLU and PubMed), **(2) frozen-feature extractor evaluation**, and **(3) additional diversity metrics** (two metrics, MAUVE and Self-BLEU), showing that DiDi-Instruct preserves the teacher model’s capability while improving efficiency.
>
> **(1) Domain Adaptation**
>
> We first evaluate downstream learning via supervised fine-tuning on Massive Multitask Language Understanding (MMLU [3]). All models (MDLM, SDTT, DiDi-Instruct) share the same 169M architecture and are trained for 5k SFT steps. DiDi-Instruct **matches or slightly exceeds** the teacher on both NLL reduction and accuracy improvement:
>
> | Model             | NLL (before) | NLL (after) $\downarrow$ | ΔNLL $\uparrow$    | ACC (before) | ACC (after) $\uparrow$ | ΔACC $\uparrow$ |
> | ----- | ---- | ---- | ---- | ------------ | ------------- | ------ |
> | MDLM              | 9.879        | 2.814         | 7.065     | 20.6%        | 25.3%         | 4.7%   |
> | SDTT              | 9.773        | 2.844         | 6.929     | 21.2%        | 24.3%         | 3.1%   |
> | **DiDi-Instruct** | 9.960        | 2.815         | **7.145** | 21.6%        | 25.1%         | 3.5%   |
>
> We further assess adaptation on the PubMed corpus with 5k SFT steps. DiDi-Instruct again closely matches the teacher and outperforms SDTT and DUO:
>
> | Model             | NLL (before) | PPL (before) | NLL (after) $\downarrow$ | PPL (after) $\downarrow$ |
> | ----------------- | ------------ | ------------ | ----------- | ------------- |
> | MDLM              | 3.761        | 42.983       | 3.217       | 24.944        |
> | SDTT              | 3.938        | 51.336       | 3.243       | 25.602        |
> | DUO               | 4.028        | 56.174       | 3.338       | 27.734        |
> | **DiDi-Instruct** | 3.879        | 47.563       | **3.215**   | **24.894**    |
>
> **Analysis:** These results confirm that **DiDi-Instruct retains strong adaptability to new domains**.
>
> **(2) Frozen Feature Extractor Evaluation**
>
> To assess representation quality independently of task-specific training, we freeze the entire backbone and train only a lightweight classification head on the MRPC dataset [4]. We report validation accuracy and F1 to assess each model’s inherent sentence-level semantic representations.
>
> | Model             | ACC $\uparrow$     | F1 $\uparrow$      |
> | ----------------- | ---------- | ---------- |
> | MDLM             | 59.6%     | 72.2%     |
> | SDTT              | 59.1%     | 72.0%     |
> | **DiDi-Instruct** | **62.3%** | **73.3%** |
>
> **Analysis:** This demonstrates that **DiDi-Instruct preserves (and slightly improves) the teacher’s sentence-level representations**.
>
> **(3) Additional Diversity Metrics**
>
> To report metrics beyond perplexity and entropy, we evaluate MAUVE (distribution similarity) and Self-BLEU (diversity) across multiple sampling budgets. DiDi-Instruct consistently achieves higher MAUVE scores and lower Self-BLEU compared to SDTT and DUO.
>
> | Metrics              | Methods       | 8     | 16    | 32    | 64    | 128   |
> |------------|-------------|-------|-------|-------|-------|-------|
> | MAUVE $\uparrow$ ($\times 10^{-3}$)     | MDLM          | 4.77  | 6.76  | 6.46  | 5.50  | 5.97  |
> |                      | SDTT          | 5.60  | 5.38  | 5.70  | 5.04  | 5.96  |
> |                      | DUO           | 5.75  | 5.15  | 5.99  | 5.85  | 6.05  |
> |                      | DiDi-Instruct | 5.84  | 6.55  | 6.54  | 6.14  | 6.31  |
> | Self-BLEU $\downarrow$ ($\times 10^{-2}$) | MDLM          | 2.91  | 4.58  | 6.36  | 7.97  | 9.33  |
> |                      | SDTT          | 5.02  | 7.98  | 10.09 | 12.49 | 12.92 |
> |                      | DUO           | 8.26  | 9.28  | 9.62  | 8.84  | 8.94  |
> |                      | DiDi-Instruct | 4.82  | 4.64  | 5.24  | 6.25  | 8.24  |
>
> **Analysis:** We attribute these improvements to **score decomposition** (Eq. 9) and **adversarial distribution-matching objective**. Unlike regression-based baselines (SDTT) that tend toward mode-seeking behavior (high Self-BLEU), score decomposition exposes the student to intermediate noisy states, effectively preserving diversity. Furthermore, adversarial training explicitly aligns the student's marginals with the teacher's distribution rather than just matching trajectories, resulting in superior distributional fidelity as reflected by higher MAUVE scores.
>
> **References**
>
> [3] Measuring Massive Multitask Language Understanding. ICLR 2021
>
> [4] Automatically Constructing a Corpus of Sentential Paraphrases. IWP 2005
>
> **We hope our rebuttal has fully addressed all of your concerns. If you still have remaining concerns, please do let us know. We are very glad to bring more clarifications!**
>
> Best regards,  authors of the submission #19443

---

> > ### Comment · Reviewer_HAtL · 2025-11-28
> >
> > Thanks the authors for their feedback. I do think the validation on different models will make the contribution more solid. Currently I have no further concerns.

---

> ### Author Response · Authors · 2025-11-28
>
> Thank you again for your thoughtful feedback and encouraging confirmation of our revisions. We will further refine and polish the manuscript to ensure the revised version is as clear, rigorous, and polished as possible.

---

### Official Review · Reviewer_Qgt2 · 2025-10-31

**Soundness:** 4
**Presentation:** 3
**Contribution:** 4
**Rating:** 8
**Confidence:** 3

**Summary:**

This paper introduces Discrete Diffusion Divergence Instruct (DiDi-Instruct), a training-based distillation method that initializes from a pre-trained dLLM to create a fast, few-step student generator. The framework is founded on integral KL-divergence minimization, which is reformulated from a policy gradient perspective to derive a tractable update rule for the discrete state space of language.

**Strengths:**

- The paper proposes a principled training method by reformulating the IKL distillation objective from a policy gradient perspective, which provides a tractable update rule for discrete spaces.
- DiDi-Instruct achieves new state-of-the-art results on the OpenWebText benchmark, consistently delivering lower PPL across 8-128 NFEs with over 20x faster distillation time than competing methods.

**Weaknesses:**

- The proposed Reward-Guided Ancestral Sampler (RGAS) introduces new hyperparameters ($h$, $M$), but there is no sensitivity analysis.
- It is unclear how the baseline results like DUO or SDTT are obtained.

**Questions:**

- Could you clarify the experimental setup for the SDTT and DUO baselines?
- Could you provide a sensitivity analysis for the RGAS hyperparameters ($h$, $M$) and elaborate on the stability of the adversarial training?

---

> ### Author Response · Authors · 2025-11-21
> **Response to Reviewer Qgt2**
>
> We appreciate your appreciation and constructive feedback. We have addressed your concerns in our revision (we highlighted paragraphs specifically related to your comments in **red**, and common to multiple reviewers in **purple**).
>
> > **Q1.** It would be great to include sensitivity analysis for RGAS.
>
> **A1.** We are sorry for the confusion. We conduct two additional experiments: **(1) Performance Landscape Analysis** and **(2) Sensitivity Analysis with fANOVA** to address this.
>
> **(1) Performance Landscape Analysis.**
>
> To visualize the joint effect of $h$ and $M$ on generation quality, we performed a dense grid search. We sweep $h$ from 0 to 100, and $M$ over the set $\{1, 2, 4, 8, 16, 32\}$. We recorded the generative PPL and Entropy for each point. Tables below show a subset of these results.
>
> *PPL Landscape at NFE=16*
>
> | $h \setminus M$ |  1 |  2 |  4 |  8 | 16 | 32 |
> | :-- | :--: | :--: | :--: | :--: | :--: | :--: |
> | 0  | 58.83  | 58.57  | 57.11  | 54.16  | 52.18  | 48.23  |
> | 20 | 40.89  | 40.32  | 42.01  | 40.06  | 40.10  | 37.09  |
> | 40 | 37.52  | 38.44  | 39.33  | 38.76  | 36.18  | 34.86  |
> | 60 | 36.87  | 36.25  | 37.36  | 36.61  | 35.72  | 33.34  |
> | 80 | 35.45  | 34.98  | 35.84  | 35.53  | 33.17  | 32.02  |
> | 100| 34.18  | 34.51  | 35.10  | 33.56  | 32.45  | 30.37  |
>
> *Entropy Landscape at NFE=16*
>
> | $h \setminus M$ | 1 | 2 | 4 | 8 | 16 | 32 |
> | :-- | :--: | :--: | :--: | :--: | :--: | :--: |
> | 0 |  5.46 |  5.46 |  5.45 |  5.44 |  5.44  |  5.44  |
> | 20 |  5.25 |  5.24 |  5.23 |  5.24 |  5.24  |  5.23  |
> | 40 |  5.12 |  5.13 |  5.13 |  5.14 |  5.12  |  5.16  |
> | 60 |  5.06 |  5.06 |  5.01 |  5.10 |  5.01  |  5.09  |
> | 80 |  5.04 |  5.04 |  4.96 |  5.00 |  4.92  |  4.98  |
> | 100 |  4.92 |  4.92 |  4.89 |  4.91 |  4.89  |  4.91  |
>
> **Conclusion**: $h$ dominates the behavior at 16 NFEs: increasing $h$ lowers PPL but also reduces entropy. Thus, good performance requires choosing a moderate $h$ to balance accuracy and diversity.
>
> **(2) Sensitivity Analysis with fANOVA.**
>
> We further conduct Global Sensitivity Analysis to formally quantify the importance of $h$ and $M$. We adopted a Latin Hypercube Sampling strategy, generating 1,000 parameter pairs with $h$ sampled continuously from $[20, 60]$ (because this range maintains a good balance between PPL and entropy) and $M$ sampled discretely from integers in $[1, 32]$. We ran this 1,000-trial experiment separately for NFE=8, 16, and 32, maintaining a constant total of 16 sampled sequences per trial. The resulting PPL data were then analyzed using Functional ANOVA [1] to decompose the output variance into contributions from each hyperparameter.
>
> | Parameter | 8 NFEs | 16 NFEs | 32 NFEs |
> | :---: | :---: | :---: | :---: |
> | $h$ | 92.7% | 90.0% | 63.7% |
> | $M$ | 7.3% | 6.0% | 36.3% |
>
> **Conclusion**: The result reveals a clear trade-off: prioritize $h$ for few-step generation, shift to $M$ as budget increases.
>
> > **Q2.** Clarify the experimental setup for the SDTT and DUO baselines
>
> **A2.** Sorry for the confusion. To ensure a fair and accurate comparison with the SDTT and DUO, we strictly followed their official GitHub repositories and experimental configurations. Specifically:
> - For SDTT, we used the same 169M teacher model pre-trained for 1024 NFEs (identical to ours) and performed 7 rounds of distillation, each with 10K steps (70K steps in total), to obtain an 8-step student model.
> - For DUO, we adopted the 169M USDM teacher model (1024 NFEs) from their official repository and conducted 5 rounds of distillation (50K total steps). We also evaluated their pre-trained few-step student model available in the repo.
>
> In our experiments, we found that the performance metrics of SDTT and DUO vary. However, to ensure the most robust and fair comparison, **we reported metrics in their published papers, which are also the best ones that we evaluated in-house.** We will add a short paragraph summarizing this setup to the Appendix to make this clearer.
>
> > **Q3.** Stability of adversarial training
>
> **A3.** We agree that adversarial-like training requires some techniques. In practice, we (1) initialize both the discriminator and the student from the pre-trained teacher, (2) freeze the student initially to warm up the discriminator, and (3) apply reward/gradient clipping to prevent exploding updates of the discriminator. During training, we also verify stability by tracking the discriminator accuracy as a stability signal. We found that with the mentioned three tricks, the discriminator training becomes stable and helpful. We will discuss this in Section 4.4
>
> **Reference**
>
> [1] An Efficient Approach for Assessing Hyperparameter Importance. ICML 2014
>
> **Dear reviewer, we hope our rebuttal has fully addressed all of your concerns. If you still have remaining concerns, please do let us know. We are very glad to bring more clarifications!**
>
> Best regards,
>
> Authors of the submission #19443

---

### Author Response · Authors · 2025-11-28
**Summary to PC SAC and AC**

Dear Program Committee, Senior Area Chairs, and Area Chairs,

We sincerely appreciate the time and effort PC, SAC, AC, and the reviewers have devoted to evaluating our submission. The constructive feedback has been incredibly valuable in helping us strengthen our work. For your convenience, we provide below **a concise summary** of the key revisions, new experiments, and clarifications we have **incorporated into the updated manuscript** to address the reviewers' comments.

---

### **1.** Strengths:

The paper received **positive ratings (8/6/6)** from the reviewers. All reviewers unanimously highlighted the **principled nature, novelty, and empirical strength** of DiDi-Instruct.

The proposed DiDi-Instruct was described as *“a mathematically rigorous solution to the difficult problem of distillation in a discrete state space”* (HAtL), *“a principled training method…which provides a tractable update rule for discrete spaces”* (Qgt2), and *“different from time-matching distillations”* while providing *“a route to few-step discrete generation”* (n4tg).

Empirically, reviewers agreed that DiDi-Instruct “*achieves new state-of-the-art results* on the OpenWebText benchmark” with “over *20× faster distillation time* than competing methods” (Qgt2) and “*consistently outperforms* other distillation methods and the GPT-2 baseline in perplexity across all sampling budgets,” while maintaining “small entropy loss” and avoiding mode collapse (HAtL, n4tg).

The clarity and structure of the manuscript were also appreciated: the paper was seen as “clearly written and well structured,” with “the proof sketch and assumptions…stated” and an “explicit link to a tractable discriminator estimator”.


### **2.** Model Scalability:

All reviewers responded very positively to our work, but they also raised concerns about scalability to large models (≥8B). We would like to clarify that **this is not a limitation**. A very recent work [1] (in Nov 2025) applies **essentially the same methodology** (on-policy distillation + adversarial training) to successfully distill **GPT-5 into Qwen-2.5-14B**.  This strongly suggests that **our core principles scale naturally to large diffusion LLMs** such as LLaDA.

We are already **actively scaling DiDi-Instruct to LLaDA-8B-Base**, and early results are consistent with this expectation. We hope our methodology provides a useful foundation for accelerating future large-scale dLLMs.

### **3.** Extended Evaluation Metrics (Reviewers HAtL and n4tg, Page 8 Sec 5.1):

We added *MAUVE* (distribution similarity) and *Self-BLEU* (diversity), which validate that DiDi-Instruct maintains strong diversity, avoids mode collapse, and outperforms the baselines (MDLM, DUO, SDTT).

### **4.** Practical Utility and Latency Analysis (Reviewer n4tg, Page 8 Sec 5.1):

We conducted a practical latency benchmark, confirming *at least* a *13.2x reduction in latency (0.432s per 1k tokens)* compared to comparable Autoregressive models (5.719s per 1k tokens) at matched perplexity.

### **5.** Downstream Performance Validation (Reviewers HAtL and n4tg, Page 8 Sec 5.3):

We added *Domain Adaptation* (MMLU, PubMed) and *Frozen Feature Extraction* (MRPC) to demonstrate that DiDi-Instruct retains the teacher’s semantic capabilities and outperforms existing few-step baselines.

### **6.** Ablation Studies of RGAS inference (Reviewer Qgt2, Page 8 Sec 5.2):

We added a performance landscape analysis and fANOVA study to show RGAS behaves *predictably and stably* across hyperparameters $h$ and $G$.

### **7.** Adversarial Training Stability (Reviewer Qgt2, Page 6 Remark 4.3):

We expanded the discussion and detailed our *warm-up, clipping, and warm-up strategy*, which makes adversarial training stable in practice.

### **8.** Protein Sequence Diversity (Reviewer HAtL, Page 10 Sec 5.5):

We added a *MMseqs2 clustering analysis* (Cluster Entropy and Average Cluster Size). Results confirm that DiDi-Instruct maintains competitive diversity, avoiding characteristic signatures of mode collapse.

### **9.** Baseline Reproducibility (Reviewer Qgt2, Page 26 Sec E.1):

We clarified the experimental setup for SDTT and DUO, strictly following their official repositories and reporting *the best results* consistent with both papers and our reproduction, ensuring fully fair comparisons.

### **10.** DiMO Baseline Clarification (Reviewer n4tg, Page 22 Sec B.3):

We clarified that DiMO is not directly applicable due to a *modality gap*, and extending it to language model distillation is non-trivial, which we leave for future work.

---

**We will continue polishing the paper and further refine the presentation to help readers capture the key ideas and contributions more clearly.** Thank you for your time and guidance throughout this process!

Sincerely,

Authors of Submission #19443

**Reference**

[1] Black-Box On-Policy Distillation of Large Language Models. arXiv:2511.10643. Nov 13 2025

---

### Meta-Review · Area_Chair_T3He · 2026-01-06

**Summary:**

This paper considers the distillation problem of discrete diffusion models. The authors propose a method termed DiDi-instruct to achieve few step generation of diffusion LLM. The method can be viewed as a discrete counterpart of score distillation for continuous diffusion models. It also introduces a discriminator to learn the density ratio between the student and teacher distributions. This discriminator can also be used to improve the inference through a reward-guided ancestral sampler (algorithm 2).

**Reviewer Concerns:**

All the reviewers agree the overall idea is interesting and the result is solid. Two main criticisms lie on the scale of the experiments and the choices of baselines. The authors use a small self-trained dLLM for evaluation while there exist many open-source large scale dLLMs. Several baselines such as DUO and DiMO could also be included to strengthen the paper. The reviewers also suggest to conduct several ablation studies. These ablation studies are added in the rebuttal.

**Reviewer Scores:**

no

---

### Decision · Program_Chairs · 2026-01-26

Accept (Poster)